# RIPK1 ablation in T cells results in spontaneous enteropathy and TNF-driven villus atrophy

Jelle Huysentruyt[1,2], Wolf Steels[1,2], Mario Ruiz Pérez [1,2], Bruno Verstraeten[1,2], Tatyana Divert[1,2], Kayleigh Flies[1,2], Kelly Lemeire[1,2], Nozomi Takahashi [1,2], Elke De Bruyn [3], Marie Joossens [3], Andrew S Brown[1,4], Bart N Lambrecht[1,4], Wim Declercq[1,2], Tom Vanden Berghe[1,2,5], Jonathan Maelfait [1,2], Peter Vandenabeele [1,2,6 ✉] & Peter Tougaard[1,2,6]

## Abstract

RIPK1 is a crucial regulator of cell survival, inflammation and cell death. Human RIPK1 deficiency leads to early-onset intestinal inflammation and peripheral T cell imbalance, though its role in αβT cell-mediated intestinal homeostasis remains unclear. In this study, we demonstrate that mice with RIPK1 ablation in conventional αβT cells (*Ripk1^ΔCD4*) developed a severe small intestinal pathology characterized by small intestinal elongation, crypt hyperplasia, and duodenum-specific villus atrophy. Using mixed bone marrow chimeras reveals a survival disadvantage of αβT cells compared to γδT cells in the small intestine. Broad-spectrum antibiotic treatment ameliorates crypt hyperplasia and prevents intestinal elongation, though villus atrophy persists. Conversely, crossing *Ripk1^ΔCD4* with TNF receptor 1 *Tnfr1^−/−* knockout mice rescues villus atrophy but not intestinal elongation. Finally, combined ablation of *Ripk1^ΔCD4* and *Casp8^ΔCD4* fully rescues intestinal pathology, revealing that αβT cell apoptosis in *Ripk1^ΔCD4* drives the enteropathy. These findings demonstrate that RIPK1-mediated survival of αβT cells is essential for proximal small intestinal homeostasis. In *Ripk1^ΔCD4* mice, the imbalanced T cell compartment drives microbiome-mediated intestinal elongation and TNF-driven villus atrophy.

**Keywords** Caspase-8; TNFR1; Duodenal Pathology; T Cell Imbalance; Intestinal Epithelial Layer
**Subject Categories** Autophagy & Cell Death; Immunology; Molecular Biology of Disease

## Introduction

Inflammatory intestinal diseases, including conditions like Crohn's disease, ulcerative colitis, and celiac disease, are complex disorders that each are characterized by significant heterogeneity in pathology across patients (Hendrickson et al, 2002; DeRoche et al, 2014; Owen and Owen, 2018). This variability complicates treatment strategies and underscores the need for a deeper understanding of the factors that drive intestinal inflammation. Given the high variability among patients, identifying specific proteins and genetic factors that influence susceptibility to intestinal inflammation is essential to understanding the mechanisms that influence intestinal pathology and may result in the identification of novel drug targets (Van Limbergen et al, 2014).

One such key factor is Receptor-Interacting Protein Kinase 1 (RIPK1), a regulator of inflammation, survival, and cell death. RIPK1's dual role in either promoting cell survival or driving inflammation and apoptosis positions it as a central element in inflammatory responses (Newton, 2020). Clinical cases show that patients with homozygous loss-of-function mutations in the *RIPK1* gene, resulting in RIPK1 deficiency, experience early-onset intestinal inflammation, underscoring RIPK1's essential role in immune regulation and its potential as a target for intervention (Li et al, 2019; Uchiyama et al, 2019; Sultan et al, 2022; Cuchet−Lourenço et al, 2018). Loss of RIPK1 in humans is further characterized as a primary immunodeficiency, resulting in peripheral T cell lymphopenia and enhanced T cell-mediated TNF and IFNγ secretion (Sultan et al, 2022; Li et al, 2019; Dai et al, 2024; Uchiyama et al, 2019). Whether RIPK1 activation results in inflammation or cell death depends on the cellular context (Rickard et al, 2014; Silke et al, 2015; Newton, 2020). Following the binding of TNF to TNFR1, RIPK1 is kept in a pro-survival mode by multiple phosphorylation- and ubiquitination-dependent checkpoints that prevent the activation of its kinase function (Delanghe et al, 2020). Inhibition of these checkpoints allows the transition of RIPK1 from a pro-survival scaffold into an active kinase, triggering TNF-mediated induction of apoptosis or necroptosis depending on the activation of caspase-8. This shift to active kinase function is critical in

[1]VIB-UGent Center for Inflammation Research, VIB, Ghent, Belgium. [2]Department of Biomedical Molecular Biology, Ghent University, Ghent, Belgium. [3]Laboratory of Microbiology, Department of Biochemistry and Microbiology, Faculty of Science, Ghent University, Ghent, Belgium. [4]Department of Internal Medicine and Pediatrics, Ghent University, Ghent, Belgium. [5]Department of Biomedical Sciences, University of Antwerp, Antwerp, Belgium. [6]These authors contributed equally as senior authors: Peter Vandenabeele, Peter Tougaard. ✉E-mail: Peter.Vandenabeele@irc.vib-ugent.be

inflammatory responses and can contribute to disease when dysregulated (Dondelinger et al, 2019; van Loo and Bertrand, 2022). Besides RIPK1 deficiency, mutation of the caspase-8-specific cleavage site (Asp325) leads to autoinflammatory disorders due to enhanced RIPK1 kinase activity-mediated apoptosis and necroptosis (Lalaoui et al, 2019; Newton et al, 2019).

Contrary to human patients with RIPK1 deficiency (Uchiyama et al, 2019; Cuchet-Lourenço et al, 2018; Li et al, 2019), whole-body RIPK1 deficiency in mice leads to perinatal death, limiting the ability to study its role across tissues (Kelliher et al, 1998). Therefore, conditional knockout models are required to investigate the cell type-specific effects of RIPK1 deficiency, as they allow for the targeted deletion of RIPK1 in specific tissues or immune cell populations, offering insights into tissue-specific functions and disease mechanisms that global knockout models cannot capture. As such, the pivotal role of RIPK1 in promoting cell survival has been demonstrated in various cell types, including skin keratinocytes (Dannappel et al, 2014) and intestinal epithelial cells (IECs) (Takahashi et al, 2014; Dannappel et al, 2014). Loss of RIPK1 in T cells leads to peripheral T cell lymphopenia with an increased proportion of CD4$^+$ and CD8$^+$ effector T cells (Huysentruyt et al, 2024; Dowling et al, 2016; Imanishi et al, 2023; Wang et al, 2023), which may impair immune surveillance and alter immune responses, particularly in tissues such as the intestines, where effector T cells play a critical role in regulating inflammation (Cheroutre et al, 2011). Thymic development of conventional T cells is largely unaffected (Webb et al, 2019; Huysentruyt et al, 2024), suggesting that RIPK1's primary role may be more prominent in peripheral immune responses rather than in thymic selection processes. We recently showed that T cell lymphopenia following RIPK1 loss is driven by TNFR1-dependent apoptosis of peripheral naive T cells (Huysentruyt et al, 2024). No substantial pathophysiological consequences of this T cell depletion were found in *Ripk1-Lck-Cre* mice (Dowling et al, 2016). However, *Ripk1-Cd4-Cre* mice (*Ripk1*$^{\Delta CD4}$) developed age-related inflammatory disorders in the lungs, kidneys and heart, as well as sarcopenia and neurodegeneration (Imanishi et al, 2023; Wang et al, 2023), suggesting that RIPK1's role in T cells is crucial for maintenance of tissue homeostasis in various tissues. Whether RIPK1-deficiency in T cells affects intestinal homeostasis in these mice has not been reported, but this is of particular interest given the intestinal pathology observed in patients with RIPK1 deficiency (Cuchet-Lourenço et al, 2018; Li et al, 2019; Sultan et al, 2022; Uchiyama et al, 2019).

Given that T cells are central to driving intestinal pathology in both Crohn's disease and celiac disease (Cheroutre et al, 2011; Meresse et al, 2004; Hu et al, 2022), it is essential to investigate the role of RIPK1 in T cells and its contribution to maintaining intestinal homeostasis. Patients with Crohn's disease can develop duodenal inflammation, although proximal small intestinal disease is mainly observed in pediatric patients (Lightner, 2018; Gupta et al, 2008). The pathology of celiac disease and autoimmune enteropathy primarily manifests in the proximal small intestine (Gentile et al, 2012; Lebwohl et al, 2018), conditions where RIPK1-mediated immune regulation could play a pivotal role in modulating inflammatory responses and maintaining intestinal homeostasis. Duodenal inflammation is often associated with lymphocytosis of intraepithelial lymphocytes (IELs) (Järvinen et al, 2003; Patterson et al, 2015; Jabri and Sollid, 2017), a process that could be influenced by RIPK1 signaling, which regulates the balance between immune activation and tissue damage in the gut.

IELs are primarily composed of T cells that are interspersed between the intestinal epithelial cells, forming one of the body's largest and most diverse T cell compartments (Olivares-Villagómez and Van Kaer, 2018). The small intestinal IEL compartment consists of several subsets of TCRαβ$^+$ and TCRγδ$^+$ T cells, collectively referred to as αβT and γδT cells, which contribute to immune regulation and intestinal homeostasis (Cheroutre et al, 2011). Their proximity to the IECs allows them to monitor the epithelial layer for signs of infection or other stress-related factors that may compromise gut integrity (Vandereyken et al, 2020). Consequently, dysregulation of the IEL compartment can contribute to the pathology observed in T cell driven diseases such as Crohn's and celiac disease (Cheroutre et al, 2011; Meresse et al, 2004; Hu et al, 2022). Therefore, the intestinal pathology observed in human RIPK1 deficiency motivated the current study on RIPK1's role in intestinal T cell-mediated tissue homeostasis.

This study shows that *Ripk1*$^{\Delta CD4}$ mice develop spontaneous small intestinal elongation and severe pathology, including crypt hyperplasia and duodenal-specific villus atrophy, indicating that RIPK1 deficiency in T cells compromises intestinal homeostasis. While duodenal enteropathy is relatively rare in animal models, it is a prevalent condition in human patients, promoting the investigation of the underlying mechanistic mediators of this pathology in mice. Here, we reveal that RIPK1 is essential in αβT cells for maintaining small intestinal homeostasis. Depletion of RIPK1 in T cells leads to a caspase-8-driven imbalance in intestinal T cell subsets, which, in turn, results in microbiome-mediated intestinal elongation and TNF-driven villus atrophy, further exacerbating the pathology.

## Results

### Selective deletion of *Ripk1* in conventional T cells results in small intestinal inflammation and elongation

T cell-specific *Ripk1*$^{\Delta CD4}$ mice were born at the expected Mendelian ratio (Fig. EV1A), but developed chronic wasting syndrome accompanied by reduced survival (Fig. 1A–C), consistent with previous reports (Dowling et al, 2016; Imanishi et al, 2023; Wang et al, 2023). *Ripk1*$^{\Delta CD4}$ mice displayed severe lymphopenia of conventional CD4$^+$ and CD8$^+$ T cells in the spleen and mesenteric lymph nodes (mLN) (Figs. 1D and EV1B). Despite the loss of conventional T cells, no difference in spleen size was observed in aged mice, while mLN were markedly enlarged (Fig. 1E), suggestive of intestinal inflammation in *Ripk1*$^{\Delta CD4}$ mice. Indeed, young *Ripk1*$^{\Delta CD4}$ mice (age 8–12 weeks) exhibited elongation of the small intestine (SI), which was exacerbated in aged mice (>6 months) (Fig. 1E,F). No difference in the length of the colon was observed, even in aged mice (Fig. EV1C,D). None of the macroscopic pathology observed in *Ripk1*$^{\Delta CD4}$ mice was present in aged heterozygous mice (*Ripk1*$^{FL/+}$ *CD4-Cre*$^{Tg/+}$) (Fig. EV1E), excluding an allele dosage effect. Moreover, the scaffold function of RIPK1 was sufficient to maintain homeostasis, as aged RIPK1 kinase-dead (*Ripk1*$^{K45A}$) mice displayed none of the above-mentioned phenotypes in terms of body weight, lymph node size, or intestinal length (Fig. EV1F,G).

Cytokine measurements revealed increased tissue concentrations of the pro-inflammatory cytokines IFN-γ, IL-17A and TNF in

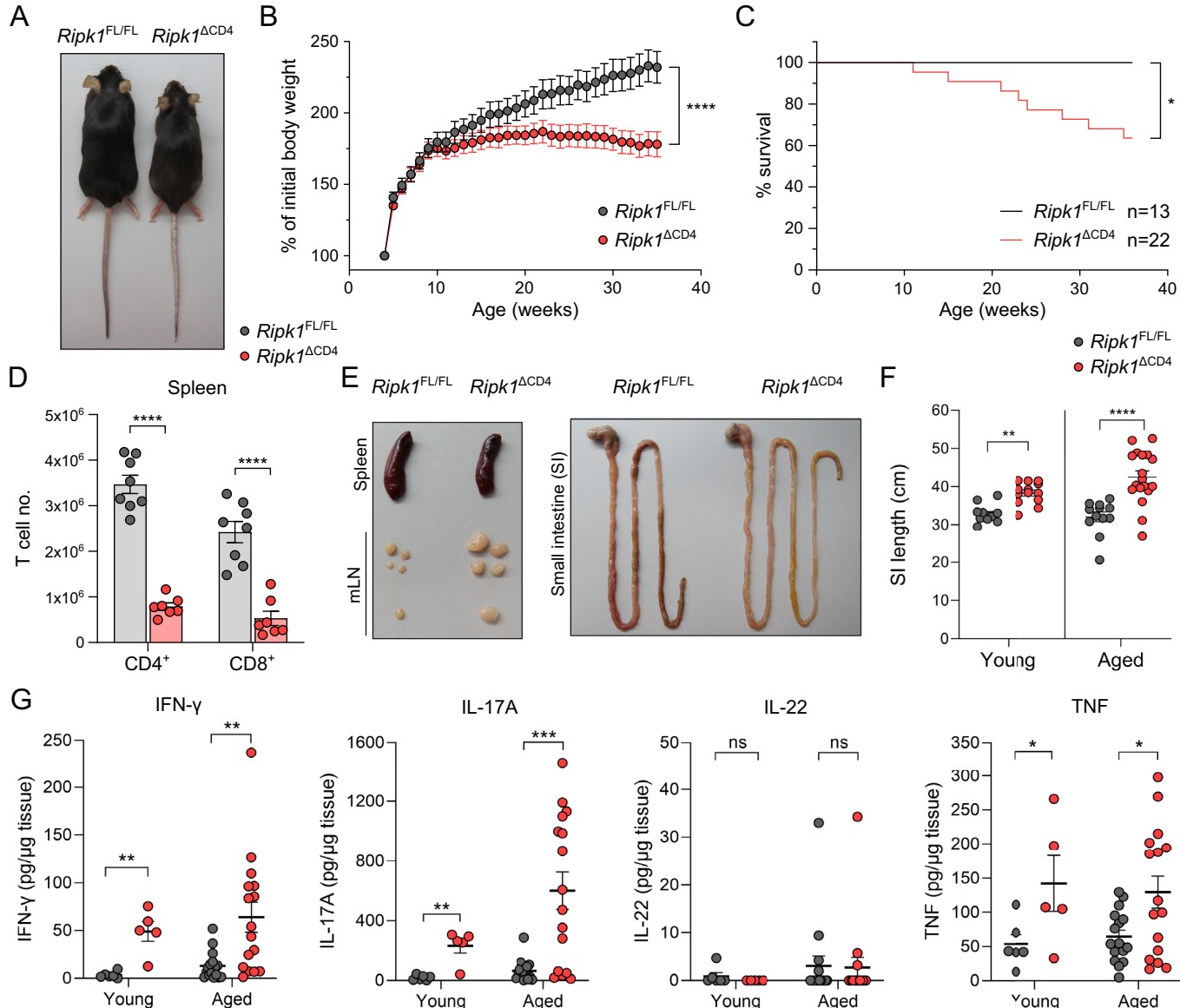

**Figure 1. Selective deletion of *Ripk1* in conventional T cells results in small intestinal inflammation and elongation.**

(A) Representative image of an aged *Ripk1*^ΔCD4^ mouse and *Ripk1*^FL/FL^ littermate. (B) Weight curves of *Ripk1*^ΔCD4^ mice (*n* = 16) and *Ripk1*^FL/FL^ littermates (*n* = 13) of both sexes, plotted as a percentage of their initial body weight at weaning (3 weeks). (C) Kaplan–Meier survival analysis comparing *Ripk1*^ΔCD4^ mice (*n* = 22) and *Ripk1*^FL/FL^ littermates (*n* = 13). (D) Absolute numbers of CD4⁺ and CD8⁺ T cells in the spleens of *Ripk1*^ΔCD4^ mice and *Ripk1*^FL/FL^ littermates, measured by flow cytometry. (E) Representative image of the spleen, mesenteric lymph nodes (mLN), and small intestine (SI) of an aged *Ripk1*^ΔCD4^ mouse and its *Ripk1*^FL/FL^ littermate. (F) Quantification of the absolute SI length in young *Ripk1*^ΔCD4^ mice (*n* = 13) and *Ripk1*^FL/FL^ littermates (*n* = 13), and aged *Ripk1*^ΔCD4^ mice (*n* = 18) and *Ripk1*^FL/FL^ littermates (*n* = 12). (G) Tissue concentrations of IFN-γ, IL-17A, IL-22, and TNF in the SI of young *Ripk1*^ΔCD4^ mice (*n* = 5) and *Ripk1*^FL/FL^ littermates (*n* = 6), and aged *Ripk1*^ΔCD4^ mice (*n* = 17) and *Ripk1*^FL/FL^ littermates (*n* = 16), were measured using multiplex assays (Meso Scale Discovery). Data are representative of three independent repeats (A, E), or are combined from two independent repeats (D), or from four independent repeats with at least two mice per group (C). Data are shown as mean ± SEM, with means being represented by bars or horizontal lines and each dot corresponding to an individual mouse. Statistical significance was calculated in Graphpad Prism for (B) by repeated-measures ANOVA (*p* < 0.0001), (C) the Gehan-Wilcoxon test for survival (*p* = 0.017), (G) two-sided unpaired T test with Welch's correction for (Young IFN-γ: *p* = 0.0038, IL-17A: *p* = 0.009, IL-22: *p* = 0.1216, and TNF: *p* = 0.0374); and (Aged IFN-γ: *p* = 0.0072, IL-17A: *p* = 0.0009, IL-22: *p* = 0.864, and TNF: *p* = 0.0183), (F) Fisher's LSD two-way ANOVA on absolute values for Young: *p* = 0.0076; and Aged: *p* < 0.0001, or (D) Log₂-transformed data for CD4⁺ and CD8⁺ T cells: *p* < 0.0001. ns = non-significant, \**p* < 0.05, \*\**p* < 0.01, \*\*\**p* < 0.001, \*\*\*\**p* < 0.0001. Young mice: 8–12 weeks old. Aged mice: >6 months old. Source data are available online for this figure.

the SI of both young and aged *Ripk1*^ΔCD4^ mice (Fig. 1G), revealing that the observed intestinal elongation is associated with enhanced type 1 and type 3 immunity. No differences were observed in the type 3 cytokine IL-22 (Fig. 1G), and levels of IL-4, IL-5, IL-6, IL-10, IL-13, and TGF-β remained unchanged or non-detectable in

young and aged mice (Fig. EV1H). In summary, our findings demonstrate that the absence of the protective RIPK1 scaffold function in conventional αβT cells leads to spontaneous SI elongation and enhanced levels of pro-inflammatory cytokines in *Ripk1*^ΔCD4^ mice.

## The duodenum of *Ripk1*^ΔCD4 mice displays villus atrophy, crypt hyperplasia, and immune cell infiltration

Histological analysis was performed on the three SI subregions (duodenum, jejunum, and ileum), isolated from aged (>6 months old) *Ripk1*^ΔCD4 mice and *Ripk1*^FL/FL littermates. Histopathological assessment by hematoxylin and eosin (H&E) staining revealed severe villus atrophy in the proximal SI (duodenum), while no villus atrophy was observed in the jejunum and ileum (Fig. 2A,B). Additionally, the intestinal crypts in the duodenum of aged *Ripk1*^ΔCD4 mice exhibited severe crypt hyperplasia, the crypt depth remained elevated throughout the small intestine but decreased toward the distal SI (Fig. 2A,C). No significant differences in duodenal villus length or crypt depth were observed in young mice (Fig. EV2A,B), and no major tissue alterations were noted in the colon of aged mice (Fig. EV2C). SEM imaging further validated duodenal villus atrophy and crypt hyperplasia, with notable morphological differences observed in villus height and crypt structure (Fig. 2D).

Enhanced proliferation of epithelial cells in the proximal SI crypts was demonstrated by Ki-67 staining (Figs. 2E,F and EV2D), suggesting a correlation with the observed SI elongation. Finally, TUNEL staining showed an enhanced presence of dead cells in the duodenum of aged mice, which was primarily observed in the epithelium (Figs. 2G and EV2E). Additionally, there was a significant increase of CD45+ immune cells in the duodenal mucosa of *Ripk1*^ΔCD4 suggesting an inflammatory response along with a markedly increased infiltration of CD45+ immune cell infiltrations in the duodenal mucosa of *Ripk1*^ΔCD4 mice suggesting, an active inflammatory response (Fig. 2G,H). In contrast, the CD45+ immune infiltrates in jejunum and ileum were less severe (Figs. 2H and EV2F). In summary, aged *Ripk1*^ΔCD4 mice develop a chronic small intestinal pathology characterized by villus atrophy, crypt hyperplasia and immune infiltration primarily affecting the duodenum. Overall, these findings suggest that *Ripk1* expression is required in conventional T cells to maintain small intestinal homeostasis.

## *Ripk1*^ΔCD4 mice have increased intra-epithelial lymphocytes and TNF producing γδT cells

Although the absolute numbers of conventional CD4+ and CD8+ T cells were profoundly decreased in the Peyer's patches (PP) of the SI of *Ripk1*^ΔCD4 mice (Fig. EV2G), TCR+ IEL numbers were significantly increased in the epithelial layer of the proximal SI and, to a lesser extent, in the lamina propria *Ripk1*^ΔCD4 mice (Fig. 3A). Conventional TCRβ+ IELs are derived from peripheral CD4+ or CD8+ T cells, whereas unconventional TCRβ+CD4⁻CD8β⁻ and TCRγδ+ subsets differentiate directly from precursor cells in the thymus (Cheroutre et al, 2011; Olivares-Villagómez and Van Kaer, 2018). Contrary to the T cell lymphopenia observed in peripheral lymphoid organs (see Figs. 1D and EV1B), conventional TCRβ+CD4+ and TCRβ+CD8β+ IEL cells, as well as TCRγδ + IEL populations, were significantly increased in *Ripk1*^ΔCD4 mice (Fig. 3B). Conversely, reduced numbers were observed in unconventional TCRβ+CD4⁻CD8β⁻ (Fig. 3B). In the lamina propria (LP), TCRβ+CD4+ LP T cell numbers remained stable, while a strong increase in TCRβ+CD8β+ and TCRγδ+ LP T cells was observed (Fig. 3C). Increased TCRγδ+ T cell numbers were also noted in the spleen, mLN and PP of *Ripk1*^ΔCD4 mice

(Figs. 3D and EV2H), showing the enhanced presence of TCRγδ+ T cells in all examined lymphoid compartments.

Mixed bone marrow chimeras were generated to investigate the intrinsic requirement of RIPK1 in IELs and LP T cells. Bone marrow from either *Ripk1*^FL/FL or *Ripk1*^ΔCD4 mice (both on CD45.2 background) were combined with an equal amount of wild-type (WT) CD45.1/2 heterozygous bone marrow cells and then injected into irradiated CD45.1 recipient mice (Fig. 3E). Twelve weeks post reconstitution, knockout/wild-type ratios of all TCRβ+ subsets were found to be reduced in the IEL and LPL compartments, while TCRγδ+ IELs and LP T cells remained unchanged (Fig. 3F,G). These findings demonstrate a cell-intrinsic survival disadvantage for αβT cells in the IEL and LPL fractions of *Ripk1*^ΔCD4 mice, while γδT cell survival remained unaffected. Since most γδT cells do not express CD4 during ontogeny (Cheroutre et al, 2011), the *Ripk1* allele in γδT cells is not affected in the *Ripk1*^ΔCD4 mice. Whether the reduced survivability of CD4+ and CD8β+ T cells influences the observed SI pathology observed in *Ripk1*^ΔCD4 mice remains unclear.

The activity of IELs and LP T cells is kept in check by regulatory T cells (Tregs) to prevent excessive inflammation at mucosal surfaces (Harrison and Powrie, 2013; Cosovanu and Neumann, 2020). FoxP3+ Tregs were found to be reduced in secondary lymphoid organs, such as the mLN (Fig. EV2I) and Peyer's patches (PP) of *Ripk1*^ΔCD4 mice (Fig. EV2J). Despite this reduction, Treg numbers were increased in the IEL of the SI, while no difference was observed in the lamina propria of *Ripk1*^ΔCD4 mice (Fig. 3H), indicating that FoxP3+ Tregs can still populate the inflamed SI of *Ripk1*^ΔCD4 mice. Intracellular expression of anti-inflammatory cytokines were measured, revealing no difference in the numbers of LP CD4+ T cells expressing TGF-β, although a marked decrease was observed in IL-10-producing CD4+ T cells (Fig. 3I,J). Intracellular staining of TNF across different T cell populations in the IEL and LP revealed enhanced TNF expression in the TCRγδ+ IELs and LPLs compared to other T cell populations in both compartments (Fig. 3K,L).

CD8αα expression is enriched in the IEL compartment on T cells and reduces TCR responsiveness (Cheroutre et al, 2011). Therefore, we investigated CD8αα+ expression on TCRγδ+ IELs. However, no difference was found in the proportion of CD8αα+ TCRγδ+ between *Ripk1*^fl/fl and *Ripk1*^ΔCD4 conditions (Fig. EV2K). Additionally, no differences were observed in the abundance of the autoimmune-associated T22 γδTCR (Lin et al, 1999) in the IEL compartment (Fig. EV2L), indicating that this specific autoimmune-associated TCR was not enriched for in the inflamed IEL compartment. Altogether, the *Ripk1*^ΔCD4 mice showed enhanced numbers of γδT cells, CD4+ and CD8β+ IELs, despite the bone marrow-mixed chimeras revealing a reduced survivability of intestinal CD4+ and CD8β+ T cells while γδT cell survival was unaffected. These differences reveal an imbalance between αβT cell and γδT cell subsets in the SI of *Ripk1*^ΔCD4 mice. Accordingly, TNF-producing TCRγδ+ T cells were highly increased in both the IEL and LPL compartments, demonstrating a pro-inflammatory T cell phenotype in the SI of *Ripk1*^ΔCD4 mice (Fig. 3M).

## *Ripk1*^ΔCD4 mice express enhanced IEC stress markers and elevated NKG2D on TCRγδ+ IELs

To obtain more in-depth information about the epithelial cells (IECs) and the IELs infiltrating the SI of *Ripk1*^ΔCD4 mice, we

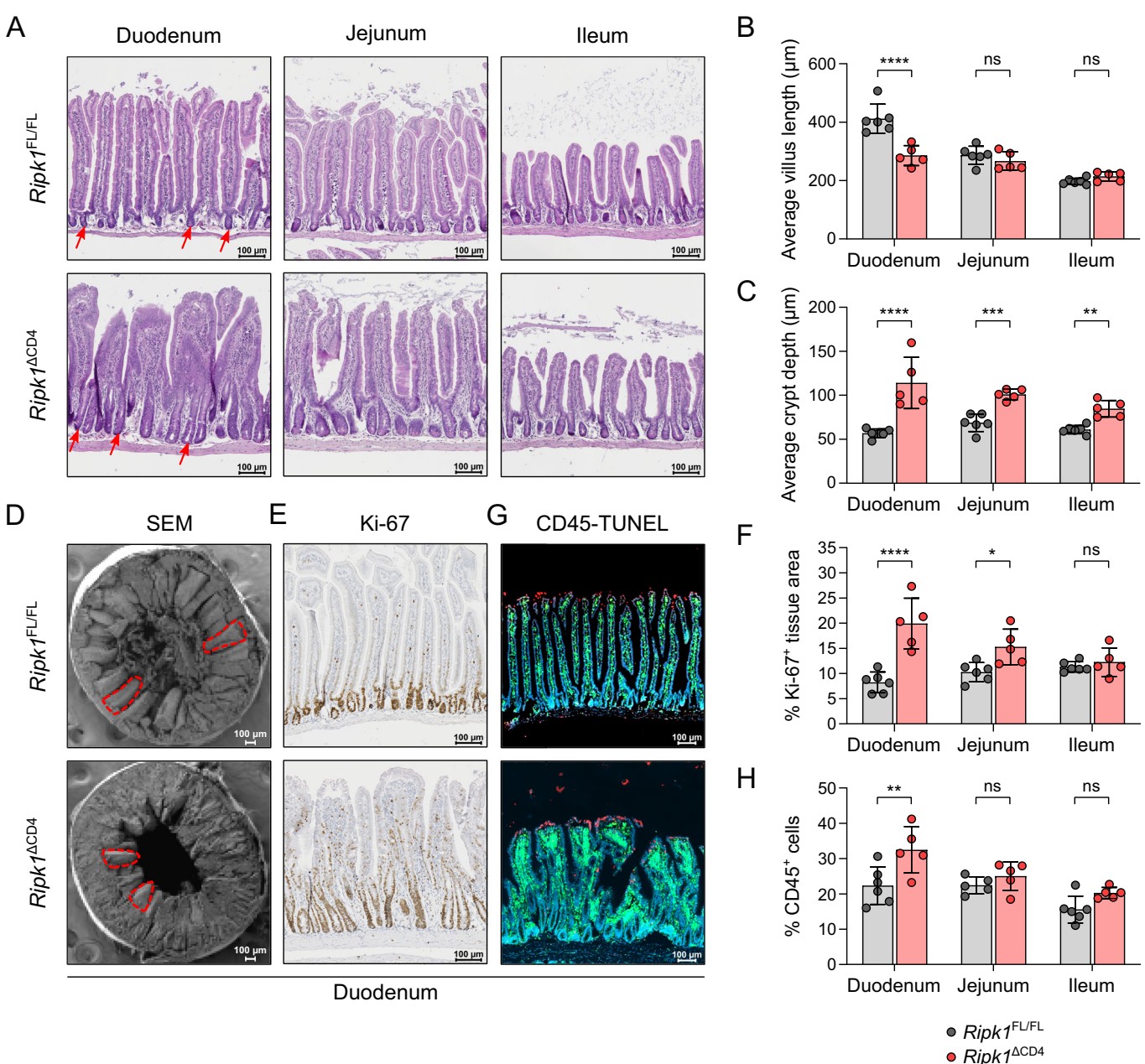

**Figure 2. The duodenum of *Ripk1*^ΔCD4 mice displays villus atrophy, crypt hyperplasia, and immune cell infiltration.**

(**A**) H&E staining was performed on sections of the duodenum, jejunum and ileum of aged *Ripk1*^ΔCD4 mice and *Ripk1*^FL/FL littermates indicating villus atrophy and crypt hyperplasia (duodenal crypts are indicated by arrows). (**B**) Quantification of the average villus length and (**C**) crypt depth in the duodenum, jejunum and ileum of aged *Ripk1*^ΔCD4 mice and *Ripk1*^FL/FL littermates. (**D**) Scanning electron microscopy (SEM) images of duodenum sections of an aged *Ripk1*^ΔCD4 mouse and *Ripk1*^FL/FL littermate. (**E**) Ki-67 staining on duodenum sections from aged *Ripk1*^ΔCD4 mice and *Ripk1*^FL/FL littermates. (**F**) Quantification of Ki-67 staining in the duodenum, jejunum, and ileum of aged *Ripk1*^ΔCD4 mice and *Ripk1*^FL/FL littermates, represented as the percentage of Ki-67-positive area to the total tissue area. (**G**) CD45-TUNEL staining on duodenum sections from aged *Ripk1*^ΔCD4 mice and *Ripk1*^FL/FL littermates (CD45 = Green; TUNEL = Red). (**H**) Quantification of CD45 staining in the duodenum, jejunum and ileum of aged *Ripk1*^ΔCD4 mice and *Ripk1*^FL/FL littermates, represented as the percentage of CD45-positive cells within the total number of cells (determined by DAPI staining). Data were obtained from (n = 5) *Ripk1*^ΔCD4 mice or (n = 6) *Ripk1*^FL/FL littermates (**B, C, F, H**), or are representative (n = 5) *Ripk1*^ΔCD4 mice and (n = 6) *Ripk1*^FL/FL littermates (**A, E, G**), or (n = 3) per group (**D**). Data are shown as mean ± SEM, with means being represented by bars and each dot corresponding to an individual mouse. Statistical significance was calculated in Graphpad Prism using Fisher's LSD two-way ANOVA on absolute values for (**B**) Duodenum: $p < 0.0001$, Jejunum: $p = 0.56$, Ileum: $p = 0.4967$; (**C**) Duodenum: $p < 0.0001$, Jejunum: $p = 0.0002$, Ileum: $p = 0.0013$; (**F**) Duodenum: $p < 0.0001$, Jejunum: $p = 0.015$, Ileum: $p = 0.9504$; and (**H**) Duodenum: $p = 0.0074$, Jejunum: $p = 0.7867$, Ileum: $p = 0.0591$. ns = non-significant, $*p < 0.05$, $**p < 0.01$, $***p < 0.001$, $****p < 0.0001$. Aged mice: >6 months. Source data are available online for this figure.

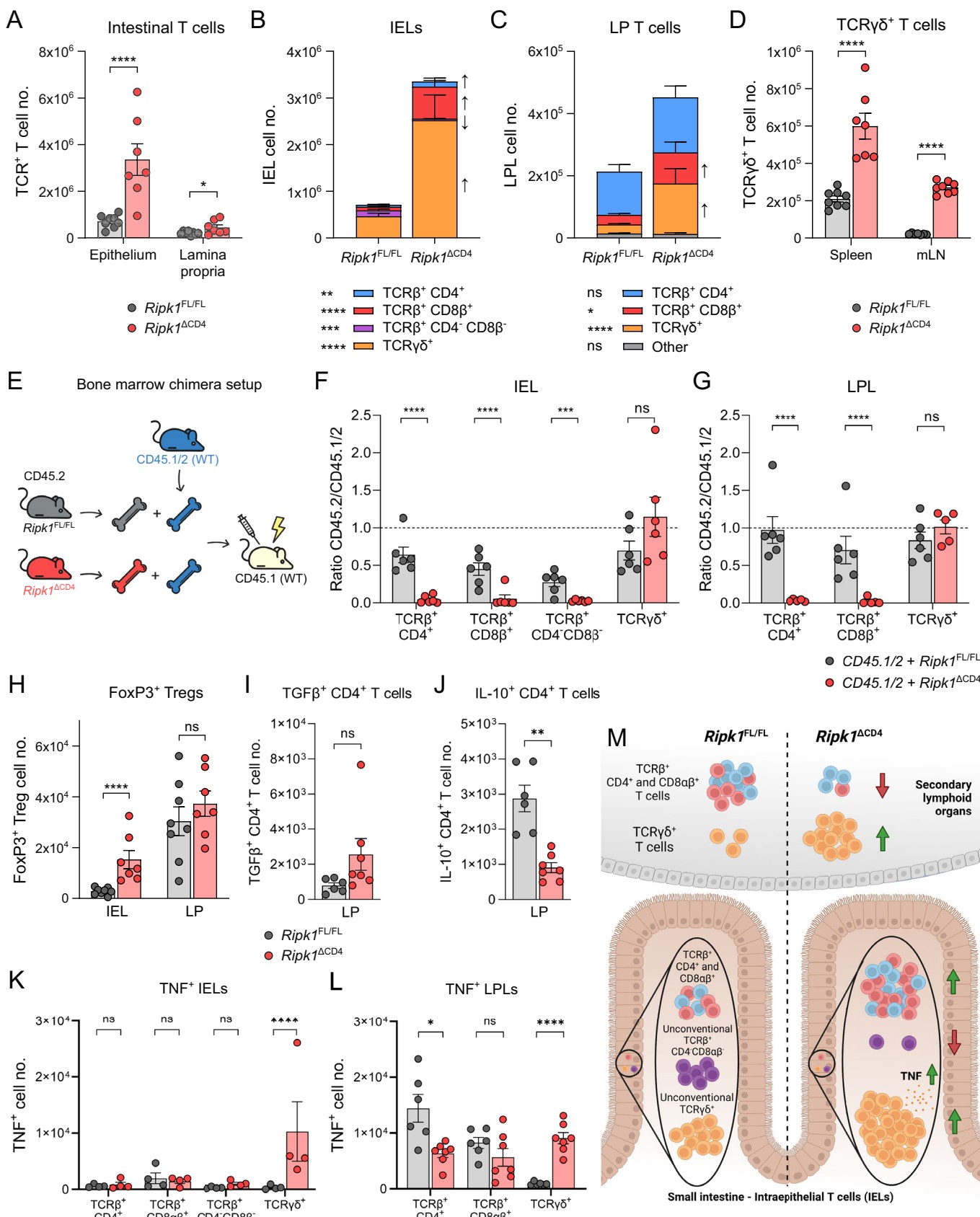

**Figure 3. *Ripk1*[ΔCD4] mice have increased intra-epithelial lymphocytes and TNF producing γδT cells.**

(A) Absolute numbers of TCR+ (both TCRβ+ and TCRγδ+) T cells within the epithelial layer and lamina propria of the small intestine (SI) in young *Ripk1*[ΔCD4] mice (n = 7) and *Ripk1*[FL/FL] littermates (n = 8), measured by flow cytometry. (B, C) Absolute numbers of SI T cell subsets in (B) IELs and (C) LP of young *Ripk1*[ΔCD4] mice (n = 7) and *Ripk1*[FL/FL] littermates (n = 8), measured by flow cytometry. (D) Absolute numbers of TCRγδ+ T cells in the spleen and mLN of young *Ripk1*[ΔCD4] mice (n = 7) and *Ripk1*[FL/FL] littermates (n = 8), measured by flow cytometry. (E) Graphical representation of the mixed bone marrow (BM) chimera experimental setup where CD45.1 recipient mice were sub-lethally irradiated and transfused with a 1:1 mixture of bone marrow cells derived from CD45.1/2 heterozygous wild-type mice and either *Ripk1*[ΔCD4] or littermate *Ripk1*[FL/FL] (both CD45.2). (F, G) Bar graphs depicting the ratios of CD45.2 to CD45.1/2T cell subsets in the IEL (F) or LP T cell (G) population isolated from the SI of mixed BM chimeras with *Ripk1*[ΔCD4] cells (n = 5), or *Ripk1*[FL/FL] cells (n = 6), measured by flow cytometry. (H) Absolute numbers of FoxP3+ Tregs isolated from the epithelial layer (IEL) and lamina propria (LP) of the SI of young *Ripk1*[ΔCD4] mice (n = 7) and *Ripk1*[FL/FL] littermates (n = 8), measured by flow cytometry. (I, J) Intracellular flow cytometry, showing the absolute numbers of CD4+ T cells in LP expressing the cytokines (I) TGF-β and (J) IL-10 obtained from (n = 7) *Ripk1*[ΔCD4] mice or (n = 6) *Ripk1*[FL/FL] littermates. Expression is shown 4 h after stimulation with cytokine stimulation cocktail. (K, L) Intracellular flow cytometry, showing the absolute numbers of TNF expressing T cell subsets in (K) IELs (n = 4) per group or (L) LP from (n = 7) *Ripk1*[ΔCD4] mice or (n = 6) *Ripk1*[FL/FL] littermates. Expression is shown 4 h after stimulation with cytokine stimulation cocktail. (M) Schematic representation of altered T cell populations in the secondary lymphoid tissues and IEL population of *Ripk1*[ΔCD4] mice compared to *Ripk1*[FL/FL] littermates. Data were combined from two independent repeats (A–L). Data are shown as mean ± sem, with means being represented by bars or horizontal lines and each dot representing an individual mouse. Statistical significance was calculated in Graphpad Prism by Fisher's LSD two-way ANOVA on Log₂-transformed data (A) IEL: $p < 0.0001$, LPL: $p = 0.0346$; (B) CD4+: $p = 0.0023$, CD8β+: $p < 0.0001$, TCRγδ+: $p < 0.0001$, TCRβ+CD4-CD8β: $p = 0.0002$; (C) CD4+: $p = 0.4729$, CD8β+: $p = 0.0232$, TCRγδ+: $p < 0.0001$, Other: $p = 0.2349$; (D) Spleen: $p < 0.0001$, mLN: $p < 0.0001$; (F) CD4+: $p < 0.0001$, CD8β+: $p < 0.0001$, TCRβ+CD4-CD8β: $p = 0.0004$ TCRγδ+: $p = 0.422$; (G) CD4+: $p < 0.0001$, CD8β+: $p < 0.0001$, TCRγδ+: $p = 0.6047$; (H) IEL: $p < 0.0001$, LPL: $p = 0.3287$; (K) CD4+: $p = 0.9996$, CD8β+: $p = 0.9971$, TCRβ+CD4-CD8β: $p = 0.2201$, TCRγδ+: $p < 0.0001$; (L) CD4+: $p = 0.0187$, CD8β+: $p = 0.0885$, TCRγδ+: $p < 0.0001$, or two-sided unpaired T test with Welch's correction (I) $p = 0.0992$; (J) $p = 0.0023$. ns = non-significant, *$p < 0.05$, **$p < 0.01$, ***$p < 0.001$, ****$p < 0.0001$. (B, C) Arrows indicate whether a population is increased (↑) or decreased (↓) in the *Ripk1*[ΔCD4] mice compared to controls. Young mice: 8–12 weeks old. IEL intraepithelial lymphocytes, LP lamina propria. Source data are available online for this figure.

performed scRNA-seq on the cells in the epithelial layer. TCRβ+ and TCRγδ+ IELs, B cells, myeloid cells and epithelial cells were identified based on cell-specific gene markers (Fig. EV3A), and depicted in an overlay Uniform Manifold Approximation and Projections (UMAP) plot (Fig. 4A). Four distinct clusters of γδT cells were identified in the IEL compartment (Fig. 4A). These clusters were classified based on their differentially expressed (DE) genes: (1) Heat-shock protein+ (HSP+), (2) Naive-like Tcf7+, (3) Cytotoxic, and (4) Ccl4+HSP- clusters (Figs. 4B,C and EV3A–C). While the abundance of HSP+ and naive-like Tcf7+ TCRγδ+ IEL clusters were lower, the cytotoxic TCRγδ+ IEL were increased in the *Ripk1*[ΔCD4] mice (Fig. 4B), which based on the overall increase in TCRγδ+ IEL indicates a phenotypic shift towards cytotoxic γδT cells. Additionally, cytotoxic TCRγδ+ IELs in *Ripk1*[ΔCD4] mice exhibited increased expression of several killer cell lectin-like receptors, such as the natural killer activating receptor gene *Klrk1* (encoding NKG2D) (Fig. 4C). Pathway analyses of the DE genes of the cytotoxic TCRγδ+ IELs further revealed highly enriched gene pathways associated with cytotoxicity, cellular senescence, and inflammatory bowel disease (IBD), among others (Fig. 4D).

A substantial increase in the numbers of NKG2D+TCRγδ+ IELs was found in the duodenum of *Ripk1*[ΔCD4] mice (Fig. 4E), validating the DE-genes findings, and suggesting enhanced cytotoxic potential upon interaction with NKG2D-ligands on stressed IECs (Hüe et al, 2004). Accordingly, DE-gene analyses of IECs revealed enhanced IEC stress response in *Ripk1*[ΔCD4] mice by the upregulation of GPX-family genes (Gpx1-2) and MGST-family genes (Mgst1-3) (Fig. 4F). These gene products are associated with protection from lipid peroxidation and reactive oxygen species (ROS) damage (Zhang et al, 2023). These findings were further underscored in the pathway analysis of the IECs, which showed a high enrichment of pathways associated with Oxidative phosphorylation/TCA cycle and ROS (Fig. 4G). Cell death-associated genes, such as caspases, intrinsic apoptosis genes, and pore-forming proteins, were further examined in the T cell subsets and epithelial cells (Fig. EV3D). While CD4+ IELs predominantly showed enhanced *Casp8* expression, the IECs displayed increased

expression of the apoptosis genes *Casp3* and *Casp7* in the IECs of the *Ripk1*[ΔCD4] mice (Fig. EV3D), consistent with the findings from the TUNEL staining (Fig. EV2E). In summary, IECs in *Ripk1*[ΔCD4] mice exhibit a transcriptome profile suggesting high levels of mitochondrial activity, ROS damage, lipid peroxidation, and apoptosis induction. Together with increased TNF-producing and NKG2D-expressing TCRγδ+ IELs, these findings suggest a possible association with the observed pathology in *Ripk1*[ΔCD4] mice.

## scRNA-seq of LP CD4+ T cells in *Ripk1*[ΔCD4] mice reveals a type 3 immune response associated with IBD

To better understand the transcriptomic landscape in the inflamed intestine, we next performed scRNA-seq on the LP compartment. We identified CD4+, CD8+ and TCRγδ+ LP T cells, in addition to innate lymphoid cells (ILCs), myeloid cells, B/plasma cells, Paneth cells, and fibroblasts (Fig. 5A), based on cell-specific gene markers (Fig. EV3E). In contrast to the IEL compartment, few DE genes were found for TCRγδ+ and the TCRβ+CD8+ LP T cells, with the exception of enhanced relative levels of ambient mRNA markers associated with plasma cell and mast cell markers (Fig. EV3F,G), similar as observed in the IELs (Fig. EV3D).

Cell death-associated genes, such as caspases, intrinsic apoptosis genes, and pore-forming proteins, were further explored in the LP-resident T cell subsets, Paneth cells, and fibroblasts (Fig. EV3H). There were minimal differences at the transcriptional level in the LP T cells (Fig. EV3H), apart from enhanced *Casp8* expression in the *Ripk1*[ΔCD4] CD4+ LP T cells, similar to those observed in the IEL fraction. Paneth cells showed enhanced *Casp1* and *Casp8* expression (Fig. EV3H), suggesting heightened caspase activity within this cell type. Pathway analysis of the TCRβ+CD4+ LP T cells indicated the strongest enrichment score for IBD and Th17 cell differentiation (Fig. 5B). We observed upregulation of *Il17a*, *Tmem176a* and *Tmem176b* in CD4+ LP T cells of *Ripk1*[ΔCD4] mice (Fig. 5C). Additionally, we found upregulation of *Tnfsf8*, the ligand for CD30 (Fig. 5C), which is expressed by activated CD4+

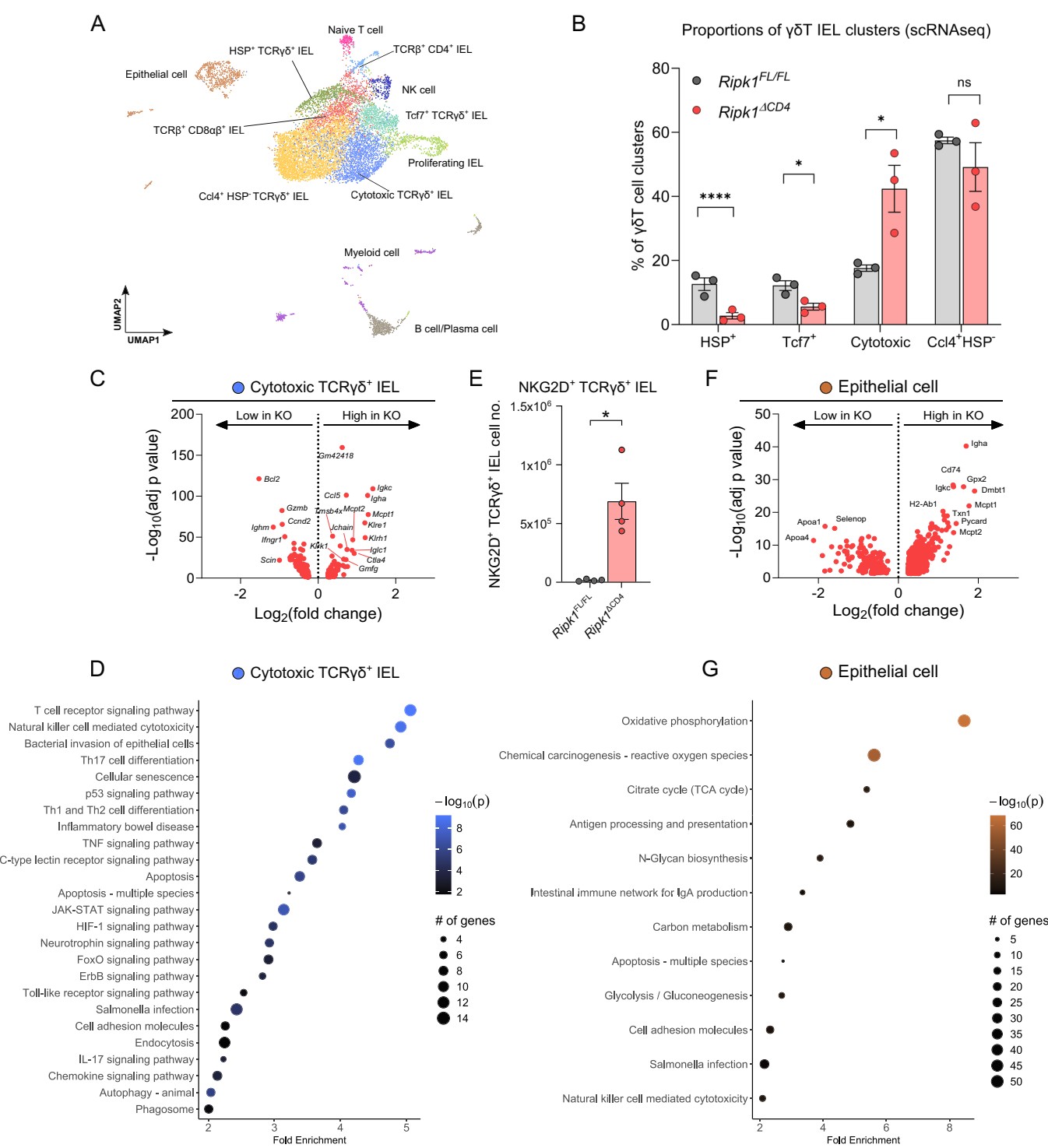

T cells and required for differentiation of both $T_h1$ and $T_h17$ cells during bacterial infections (Tang et al, 2008; Sun et al, 2010). The transcriptional upregulation of *Il17a* correlates with our protein data, showing increased tissue levels of IL-17A in the SI of *Ripk1*$^{\Delta CD4}$ mice (Fig. 1G). These findings indicate an active type-3 immune response within the SI lamina propria of *Ripk1*$^{\Delta CD4}$ mice.

## SI elongation in *Ripk1*$^{\Delta CD4}$ mice is a response to the intestinal microbiome, while villus atrophy and TCRγδ⁺ IEL expansion are not

Type 3 immunity is primarily an immune response towards extracellular bacteria and fungi (Annunziato et al, 2015). Immunoglobulin A (IgA) mediates protective immunity to the gut

**Figure 4.** *Ripk1*$^{\Delta CD4}$ **mice express enhanced IEC stress markers and elevated NKG2D on TCRγδ$^+$ IELs.**

Cells were isolated from the epithelial layer (IEL) of young *Ripk1*$^{\Delta CD4}$ *and Ripk1*$^{FL/FL}$ littermates and subjected to single-cell RNA sequencing (scRNA-seq). (A) Cells from both IEL samples were combined, clusters of TCRβ$^+$CD4$^+$, TCRβ$^+$CD8$^+$ and TCRγδ$^+$ IELs, proliferating cells, B cells and plasma cells, NK cells, myeloid cells and epithelial cells were identified and presented in a Uniform Manifold Approximation and Projection (UMAP) plot. (B) Bar plots showing the proportions of the four identified TCRγδ$^+$ IEL clusters (shown in A) comparing *Ripk1*$^{\Delta CD4}$ and *Ripk1*$^{FL/FL}$ littermate controls. (C) Volcano plot representing significant differentially expressed (DE) genes within the Cytotoxic TCRγδ$^+$ IEL cluster of *Ripk1*$^{\Delta CD4}$ mice compared to *Ripk1*$^{FL/FL}$ littermates. Significance ($-Log_{10}$ of the *p*-value) is indicated on the y-axis and $Log_2$ of the fold change in gene expression is indicated on the x-axis. (D) Pathway analysis (from PathfindeR) of DE genes in the Cytotoxic TCRγδ$^+$ IEL cluster displaying pathway Fold enrichment, statistical significance ($-Log_{10}$ of the *p*-value), and the number of DE genes in the specific pathway. (E) Absolute numbers of NKG2D$^+$ TCRγδ$^+$ IELs within the duodenum of *Ripk1*$^{\Delta CD4}$ mice and *Ripk1*$^{FL/FL}$ littermates, measured by flow cytometry. (F) Volcano plot representing significant differentially expressed (DE) genes in the Epithelial cell cluster of *Ripk1*$^{\Delta CD4}$ mice compared to *Ripk1*$^{FL/FL}$ littermates. Significance ($-Log_{10}$ of the *p*-value) is indicated on the y-axis, and $Log_2$ of the fold change in gene expression is indicated on the x-axis. (G) Pathway analysis (from PathfindeR) of DE genes from the Epithelial cell cluster displaying pathway Fold enrichment, their significance ($-Log_{10}$ of the *p*-value), and the number of DE genes in the specific pathway. Data were obtained from (*n* = 3) mice per group (A–D, F, G) or (*n* = 4) mice per group (E). (B, E) Data are shown as mean ± SEM, with means being represented by bars and each dot representing an individual mouse. Statistical significance for (B) and (E) were calculated in Graphpad Prism by Fisher's LSD two-way ANOVA on $Log_2$-transformed data (B) HSP$^+$: *p* < 0.0001; Tcf7$^+$: *p* = 0.0329; Cytotoxic: *p* = 0.0228; Ccl4$^-$HSP$^-$: *p* = 0.942, or (E) by two-sided unpaired T test with Welch's correction (*p* = 0.0222). (C, F) Wilcox test was used to determine DE genes and *p*-values were adjusted using Bonferroni correction. (D, G) Hypergeometric-distribution-based tests were used to determine significant KEGG terms and *p*-values were adjusted using Bonferroni correction. ns = non-significant, *\*p* < 0.05, *\*\*\*\*p* < 0.0001. Young mice: 8–12 weeks old. IEL intra-epithelial T lymphocytes. Source data are available online for this figure.

microbiota, contributing to intestinal homeostasis (Bunker et al, 2015). Measurements of serum IgA revealed high levels of IgA in aged *Ripk1*$^{\Delta CD4}$ mice (Fig. EV4A), suggestive of an anti-microbial response in the gut (Bunker and Bendelac, 2018). To explore whether the observed pathology in *Ripk1*$^{\Delta CD4}$ mice is a response towards the intestinal microbiome, pregnant mothers were subjected to a cocktail of broad-spectrum antibiotics in drinking water and continued treatment of their offspring for seven months (Fig. 6A). Caecum swelling was observed post-antibiotic treatment in both *Ripk1*$^{\Delta CD4}$ mice and *Ripk1*$^{FL/FL}$ littermates, indicating drastic alterations in the intestinal microbiome composition (Fig. EV4B).

Antibiotic treatment fully reversed SI elongation, and the associated epithelial Ki-67 expression and crypt hyperplasia were greatly reduced (Figs. 6B,C and EV4C). However, the antibiotics did not prevent enhanced mortality in *Ripk1*$^{\Delta CD4}$ mice (Fig. EV4D) or the lymphadenopathy of the mLN (Fig. 6D). Additionally, although the crypt hyperplasia was markedly reduced, the villus atrophy remained in the SI of aged *Ripk1*$^{\Delta CD4}$ mice on antibiotics (Fig. 6E–H). Furthermore, while the enhanced numbers of TCRγδ$^+$ IELs were unaffected by antibiotics treatment, the numbers of FoxP3$^+$ Treg, TCRβ$^+$CD8β$^{+,}$ and TCRβ$^+$CD4$^+$ IELs were reduced in the SI of antibiotics-treated *Ripk1*$^{\Delta CD4}$ mice (Figs. 6I,J and EV4E). These findings suggest that the SI elongation, the epithelial hyperproliferation, FoxP3$^+$ Treg IEL infiltration and the crypt hyperplasia in *Ripk1*$^{\Delta CD4}$ mice most likely are a response to the gut microbiome. In summary, the SI elongation, crypt hyperplasia and infiltration of Tregs, TCRβ$^+$CD4$^+$ and TCRβ$^+$CD8β$^+$ IELs within the epithelial layer appear to be microbiome-dependent, while duodenal villus atrophy and TCRγδ$^+$ IEL lymphocytosis remain unaffected by microbiome alterations.

## Villus atrophy is TNFR1-dependent, and T cell ablation of caspase-8 prevents the development of intestinal pathology

We have previously shown that peripheral naive T cells require RIPK1 to evade caspase-8-mediated apoptosis (Huysentruyt et al, 2024). Despite the enhanced levels of TCRβ$^+$ IELs, the mixed bone marrow chimeras revealed a cell-intrinsic survival disadvantage of RIPK1-deficient TCRβ$^+$ IEL and LP populations (Fig. 3F,G). Therefore, we generated mice with the *Cd4*-specfic ablation of

both *Ripk1* and *Casp8* genes (denoted as *Ripk1*$^{\Delta CD4}$*Casp8*$^{\Delta CD4}$), to determine whether blocking Caspase-8-mediated apoptosis in T cells would influence the inflammatory phenotype. Interestingly, aged (>7 months) *Ripk1*$^{\Delta CD4}$*Casp8*$^{\Delta CD4}$ mice were indistinguishable from *Ripk1*$^{FL/FL}$*Casp8*$^{FL/FL}$ littermates (Fig. 7A), displayed no lymphadenopathy of the mLN (Fig. 7B) or SI elongation in contrast to *Ripk1*$^{\Delta CD4}$ mice (Figs. 7C and EV5A).

Additionally, young *Ripk1*$^{\Delta CD4}$*Casp8*$^{\Delta CD4}$ mice showed no lymphopenia of peripheral CD4$^+$ T cells (Figs. 7D), confirming our earlier findings that RIPK1 is required for the protection of peripheral CD4$^+$ T cells from apoptosis (Huysentruyt et al, 2024). Immunohistochemical analysis of the duodenum in *Ripk1*$^{\Delta CD4}$*Casp8*$^{\Delta CD4}$ mice showed no evidence of crypt hyperplasia, villus atrophy, epithelial hyperproliferation or immune cell infiltration (Figs. 7E–H and EV5B–D). These findings demonstrate that the duodenal pathology of *Ripk1*$^{\Delta CD4}$ mice was resolved by additional T cell-specific deletion of caspase-8. Indeed, numbers of both TCRβ$^+$ IELs and TCRγδ$^+$ IELs in *Ripk1*$^{\Delta CD4}$*Casp8*$^{\Delta CD4}$ mice were comparable to those in *Ripk1*$^{FL/FL}$*Casp8*$^{FL/FL}$ littermates (Figs. 7G and EV5E). Similarly, NKG2D$^+$ TCRγδ$^+$ IELs numbers were consistent between *Ripk1*$^{\Delta CD4}$*Casp8*$^{\Delta CD4}$ mice and their littermate controls (Fig. EV5F). Moreover, the peripheral TCRγδ$^+$ T cell expansion observed in *Ripk1*$^{\Delta CD4}$ mice was prevented by additional deletion of the *Casp8* gene (Figs. EV5G and 3D). Western blot analysis revealed that neither RIPK1 nor Caspase-8 expression was reduced in the TCRγδ$^+$ T cells in any of the genotypes (Fig. EV5H), demonstrating the specificity of the *Cd4-Cre*-specific gene ablation.

Collectively, these findings indicate that the sustained expression of RIPK1 in TCRγδ$^+$ T cells likely gives them a competitive survival advantage over TCRβ$^+$ cells in the *Ripk1*$^{\Delta CD4}$ mice, resulting in enhanced presence of TCRγδ$^+$ T cells across tissues. Additionally, we found that unlike in *Ripk1*$^{\Delta CD4}$ mice, in *Ripk1*$^{\Delta CD4}$*Casp8*$^{\Delta CD4}$ mice the tissue levels of TNF, IFN-γ, and IL-17A in the SI, as well as serum levels of IgA, were comparable to those in *Ripk1*$^{FL/FL}$*Casp8*$^{FL/FL}$ littermates (Figs. 7J and EV5I–K), showing a complete rescue of the intestinal inflammation by *Cd4*-specific genetic ablation of *Casp8*. Finally, given that IEL-produced TNF can induce enhanced IEC shedding, leading to villus blunting in areas with active Crohn's disease (Hu et al, 2022), we crossed the *Ripk1*$^{\Delta CD4}$ mice to a *Tnfr1*$^{-/-}$ mouse to investigate its role in the intestinal pathology.

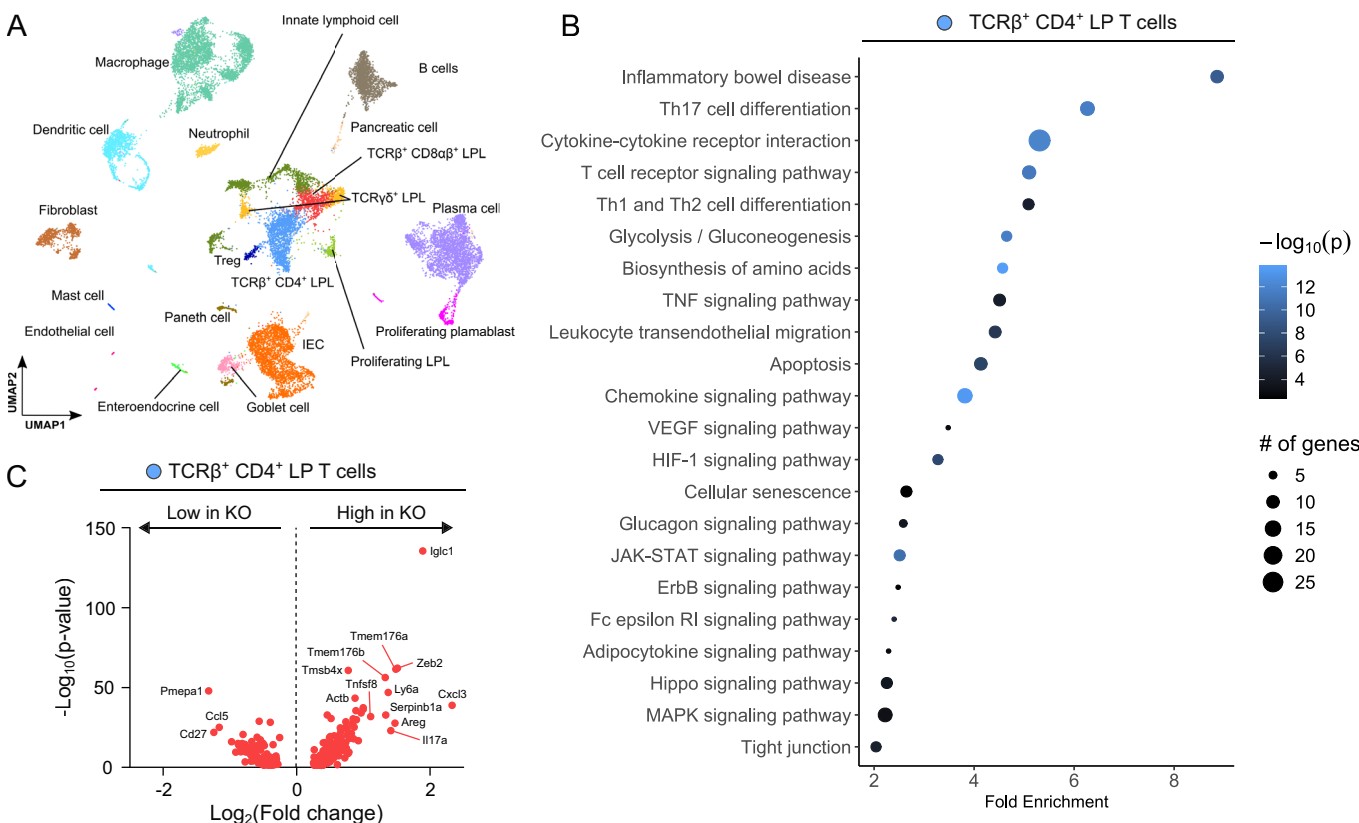

**Figure 5. scRNA-seq of LP CD4+ T cells in Ripk1ΔCD4 mice reveals a type 3 immune response associated with IBD.**

Cells were isolated from the lamina propria (LP) of young *Ripk1ΔCD4 and Ripk1FL/FL* littermates and analyzed using single-cell RNA sequencing (scRNA-seq). (A) Cells from both samples were combined, and clusters of TCRβ+CD4+, TCRβ+CD8+ and TCRγδ+ LP T cells, proliferating cells, innate lymphoid cells, myeloid cells, macrophages, B cells and plasma cells, epithelial cells and mesenchymal cells were identified and presented in a Uniform Manifold Approximation and Projection (UMAP) plot. (B) Volcano plot representing differentially expressed (DE) genes in the cluster of TCRβ+CD4+ LP T cells of *Ripk1ΔCD4* mice compared to *Ripk1FL/FL* littermates. Significance (−Log10 of the *p*-value) is indicated on the y-axis, and Log2 of the fold change in gene expression is indicated on the x-axis. (C) Pathway analysis (from PathfindeR) of DE genes in the TCRβ+CD4+ LP T cell cluster displaying pathway Fold enrichment, the significance (−Log10 of the *p*-value), and number of DE genes in the specific pathway. (B) Wilcox test was used to determine DE genes and *p*-values were adjusted using Bonferroni correction. (C) Hypergeometric-distribution-based tests were used to determine significant KEGG terms and *p*-values were adjusted using Bonferroni correction. Data were obtained from (*n* = 3) mice per group. Young mice: 8–12 weeks old. LP Lamina propria. Source data are available online for this figure.

Our findings demonstrate that while the SI length and crypt hyperplasia occur independent of TNFR1, villus atrophy was completely rescued in aged *Ripk1ΔCD4Tnfr1−/−* mice (Fig. 7K–M). In conclusion, these findings suggest that the enhanced Caspase-8-mediated apoptosis of TCRβ+ T cells drives intestinal pathology in *Ripk1ΔCD4* mice. Furthermore, while the SI elongation and crypt hyperplasia in *Ripk1ΔCD4* mice was TNFR1 independent, the villus atrophy was rescued in *Tnfr1−/−* mice, revealing that TNF mediates this pathological feature.

## Discussion

Previous studies have shown that genetic ablation of *Ripk1* in mice results in enhanced peripheral T cell apoptosis, highlighting RIPK1's important role in T cell survival (Dowling et al, 2016; Webb et al, 2019; Huysentruyt et al, 2024; Wang et al, 2023). Our findings demonstrate that the expression of *Ripk1* in conventional T cells is essential for maintaining intestinal homeostasis, underscoring its protective role in

intestinal pathologies. *Ripk1ΔCD4* mice exhibit villus atrophy and crypt hyperplasia in the duodenum, associated with elongation of the SI. Additionally, we observe an increased infiltration of T cells in the IEL and LP compartments of the SI. In *Ripk1ΔCD4* mice, certain features, such as crypt hyperplasia and intestinal elongation, were microbiota-dependent, whereas villus atrophy persisted despite broad-spectrum antibiotic treatment. The duodenal crypt hyperplasia, the intestinal elongation, and the expansion of Tregs, CD4+ and CD8β+ TCRβ+ IELs were all rescued following antibiotics, suggesting that these phenotypes represent a response towards the intestinal microbiome. The expansion of TCRγδ+ IELs and the villus atrophy remained in antibiotics-treated *Ripk1ΔCD4* mice, indicating that these symptoms were independent of the microbiome alterations. Comparing germ-free and specific pathogen-free (SPF) mice has also previously shown that the total TCRγδ+ IEL numbers are relatively unaffected by the presence of the microbiota (Bandeira et al, 1990; Hoytema van Konijnenburg et al, 2017). TCRγδ+ IELs may affect the microbial composition through the expression of anti-microbial peptides (Rezende et al, 2023). However, since the antibiotic treatment did

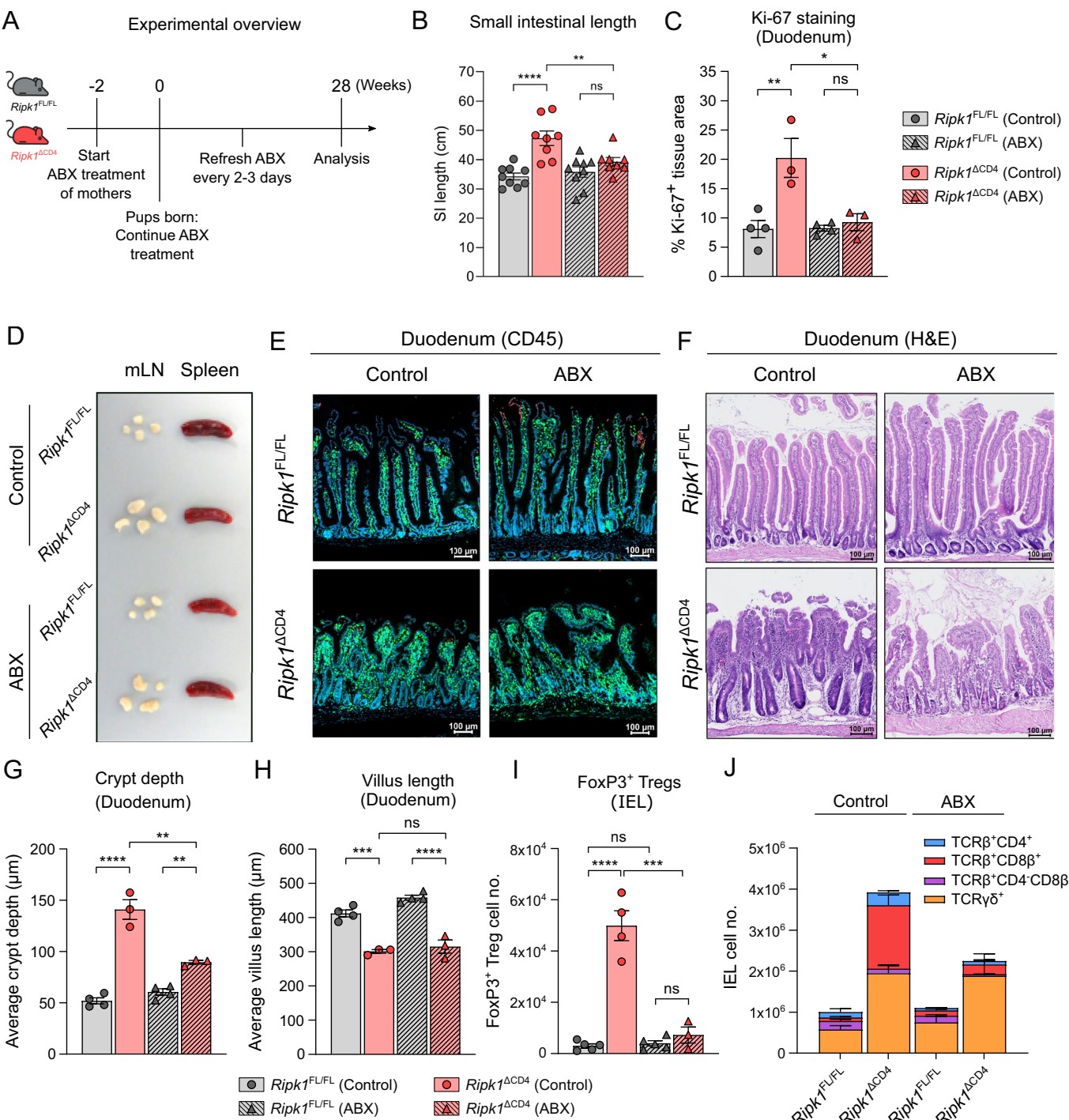

not affect the villus atrophy or increased mortality in the $Ripk1^{\Delta CD4}$ mice, the microbiota is unlikely to be the underlying cause of these pathologies.

In both humans and mice, the numbers of TCRγδ⁺ IELs are most prominent in the proximal part of the SI, and their abundance decreases down the length of the intestine (Camerini et al, 1993; Lundqvist et al, 1995; Hoytema van Konijnenburg et al, 2017). This aligns with our observation that intestinal pathology in $Ripk1^{\Delta CD4}$ mice mainly manifested in the proximal SI. We found that the TCRγδ⁺ IELs

in $Ripk1^{\Delta CD4}$ mice expressed elevated levels of NKG2D, a key activating receptor on IELs during infection and inflammation (Zhou et al, 2007; Li et al, 2017). This elevation suggests an enhanced activation state of the TCRγδ⁺ IELs in $Ripk1^{\Delta CD4}$ mice. Furthermore, TCRγδ⁺ T cell numbers are elevated in all examined peripheral tissues in $Ripk1^{\Delta CD4}$ mice. This increase in TCRγδ⁺ T cells may reflect a compensatory mechanism resulting in RIPK1-sufficient TCRγδ⁺ T cells filling the niche instead of TCRβ⁺ T cells due to the survival disadvantage of $Ripk1^{\Delta CD4}$ TCRβ⁺ T cells observed in bone marrow chimeras.

Figure 6.  SI elongation in *Ripk1*^ΔCD4 mice is a response to the intestinal microbiome, while villus atrophy and TCRγδ^+ IEL expansion are not.

(A) Schematic overview of the long-term antibiotics (ABX) treatment assay. (B) Quantification of the absolute small intestine (SI) length in ABX-treated and untreated aged *Ripk1*^ΔCD4 mice and *Ripk1*^FL/FL littermates. (C) Quantifications of Ki-67 staining in the duodenum of aged *Ripk1*^ΔCD4 mice and *Ripk1*^FL/FL littermates, with and without ABX treatment. (E, F) Duodenum sections of aged *Ripk1*^ΔCD4 mice and *Ripk1*^FL/FL littermates, with and without ABX treatment, showing (E) CD45 (green) and TUNEL (red) staining and (F) H&E staining. (E–G) Quantifications of (G) the average crypt depth and (H) the average villus length in the duodenum of aged *Ripk1*^ΔCD4 mice and *Ripk1*^FL/FL littermates, with and without ABX treatment. (I) Absolute numbers of FoxP3^+ Tregs isolated from the SI epithelial layer (IEL) of aged *Ripk1*^ΔCD4 mice and *Ripk1*^FL/FL littermates, with and without ABX treatment, as measured by flow cytometry. (J) Absolute numbers of TCRβ^+CD4^+, TCRβ^+CD8β^+, TCRβ^+CD4^-CD8β^- and TCRγδ^+ IELs of aged *Ripk1*^ΔCD4 mice and *Ripk1*^FL/FL littermates, with and without ABX treatment, measured by flow cytometry. (B) Data were combined from two independent repeats with ($n = 8$) *Ripk1*^ΔCD4 mice (both ABX and control) or ($n = 9$) *Ripk1*^FL/FL littermates (both ABX and control). (C–G) Representative of two independent repeats with ($n = 3$) *Ripk1*^ΔCD4 mice (both ABX and control groups) or ($n = 4$) *Ripk1*^FL/FL littermates (both ABX and control groups), or (I, J) Data are representative of two independent repeats with ($n = 4$) *Ripk1*^ΔCD4 mice (control) ($n = 3$) *Ripk1*^ΔCD4 mice (ABX) or ($n = 5$) *Ripk1*^FL/FL littermates (both ABX and control groups)). Data are shown as mean ± SEM, with means represented by bars or horizontal lines and each dot representing an individual mouse. Statistical significance was calculated in Graphpad Prism by two-way ANOVA on absolute values (B) *Ripk1*^FL/FL vs *Ripk1*^ΔCD4 (Control): $p < 0.0001$, *Ripk1*^FL/FL vs *Ripk1*^ΔCD4 (ABX): $p = 0.1698$, Control vs ABX (*Ripk1*^ΔCD4): $p = 0.0039$, or Fisher's LSD one-way ANOVA on Log$_2$-transformed values (C) *Ripk1*^FL/FL vs *Ripk1*^ΔCD4 (Control): $p = 0.0081$, *Ripk1*^FL/FL vs *Ripk1*^ΔCD4 (ABX): $p = 0.9743$, Control vs ABX (*Ripk1*^ΔCD4): $p = 0.0356$; (G) *Ripk1*^FL/FL vs *Ripk1*^ΔCD4 (Control): $p < 0.0001$, *Ripk1*^FL/FL vs *Ripk1*^ΔCD4 (ABX): $p = 0.0022$, Control vs ABX (*Ripk1*^ΔCD4): $p = 0.0014$; (H) *Ripk1*^FL/FL vs *Ripk1*^ΔCD4 (Control): $p = 0.0002$, *Ripk1*^FL/FL vs *Ripk1*^ΔCD4 (ABX): $p < 0.0001$, Control vs ABX (*Ripk1*^ΔCD4): $p = 0.8082$; and (I) *Ripk1*^FL/FL vs *Ripk1*^ΔCD4 (Control): $p = 0.472$, *Ripk1*^FL/FL vs *Ripk1*^ΔCD4 (ABX): $p = 0.0007$, Control vs ABX (*Ripk1*^ΔCD4): $p = 0.3538$, or (J) two-way ANOVA on Log$_2$-transformed data with the corresponding statistical analysis depicted for each population in Fig. EV4E. ns = non-significant, *$p < 0.05$, **$p < 0.01$, ***$p < 0.001$, ****$p < 0.0001$. Aged mice: >6 months old. IEL intra-epithelial T lymphocytes. Source data are available online for this figure.

In the LP, Foxp3^+ Treg numbers were similar between *Ripk1*^ΔCD4 and *Ripk1*^fl/fl genotypes. However, the number of IL-10-producing CD4 cells was decreased in the LP in the *Ripk1*^ΔCD4 mice, which suggests a reduced capacity to regulate the production of pro-inflammatory cytokines (Kessler et al, 2017; Costes et al, 2019; Brockmann et al, 2018). Accordingly, enhanced levels of IL-17A, IFNγ and TNF were observed in the SI tissue. Additionally, pathway analyses of CD4^+ LP T cells scored highest for pathways associated with IBD and Th17 differentiation, especially based on upregulation of the type-3 immune response genes, such as *Il17a*, *Tmem176a*, *Tmem176b*, and *Tnfsf8* (Ciofani et al, 2012; Drujont et al, 2016; Sun et al, 2010).

The absence of RIPK1 in murine intestinal epithelial cells (IEC) induces severe inflammatory symptoms in the gut and premature death due to excessive IEC apoptosis (Takahashi et al, 2014; Dannappel et al, 2014). Furthermore, while antibiotic treatment in *Ripk1*^ΔIEC mice rescued the intestinal pathology (Takahashi et al, 2014), a different study reported that antibiotic treatment in tamoxifen-inducible *Ripk1*^ERT2-IEC mice was unable to alleviate the pathology (Dannappel et al, 2014). This discrepancy may be due to differences in animal house conditions. Here, we demonstrate that RIPK1 deficiency in conventional T cells leads to a complex enteropathy driven by both microbiome-dependent and independent pathology.

Interestingly, duodenal pathology and villus atrophy, which have been reported in human RIPK1 deficiency (Uchiyama et al, 2019), are similar to the pathology observed in our study. This parallel underscores the potential relevance of our findings to human disease. Additionally, multiple other intestinal enteropathies can also lead to duodenal pathologies, such as Crohn's disease (Vyhlidal et al, 2021), celiac disease (Lebwohl et al, 2018), and autoimmune enteropathy (Gentile et al, 2012), and have been associated with intraepithelial lymphocytosis (Vyhlidal et al, 2021; Patterson et al, 2015; Savilahti et al, 1990; Järvinen et al, 2003; Gentile et al, 2012). In celiac disease, for instance, villus atrophy is directly associated with the activity of infiltrating IELs (Meresse et al, 2004). Furthermore, in Crohn's disease, TCRγδ^+ IELs promote pathology by inducing TNF-mediated shedding of IECs, leading to villus atrophy in areas with active disease (Hu et al,

2022). Given the increased number of TNF-producing TCRγδ^+ IELs and enhanced tissue concentrations of TNF, we crossed the *Ripk1*^ΔCD4 mice onto a *Tnfr1*^−/− background. *Ripk1*^ΔCD4*Tnfr1*^−/− mice were protected from villus atrophy, yet they still exhibited microbiome-mediated intestinal elongation and crypt hyperplasia. This suggests that TNF signaling plays a critical role in mediating villus atrophy, while other pathways remain active in tissue remodeling.

Finally, we found that the additional deletion of the *Casp8* gene in *Ripk1*^ΔCD4*Casp8*^ΔCD4 mice completely rescues the intestinal pathology observed in *Ripk1*^ΔCD4 mice. This rescue shows that apoptosis of TCRβ^+ T cells is the underlying cause of the crypt hyperplasia, villus atrophy and lymphocytosis of IELs. The complete absence of pathology in *Ripk1*^ΔCD4*Casp8*^ΔCD4 mice indicates that intestinal homeostasis relies on RIPK1-mediated survival of peripheral CD4^+ and CD8^+ T cells by preventing caspase-8-mediated apoptosis. Additionally, using *Ripk1*^K45A mice, we found that the scaffold function of RIPK1, rather than its kinase activity is crucial for T cell survival and maintenance of intestinal homeostasis.

Previous work has shown that RIPK1-deficient T cells undergo Caspase-8-mediated apoptosis (Huysentruyt et al, 2024), and other studies have reported that caspase-8 limits RIPK1-mediated necroptosis following TCR stimulation (Salmena et al, 2003; Ch'en et al, 2008). Consequently, mice and patients with mutations in the caspase-8 cleavage site of RIPK1 develop an autoinflammatory syndrome characterized by excessive RIPK1-dependent cell death (Newton et al, 2019; Lalaoui et al, 2019). Interestingly, although T cell-specific *Casp8* knockout mice also developed a wasting syndrome with reduced survival rates, the specific ablation of *Casp8* in T cells does not result in the same mucosal-associated pathology as observed in *Ripk1*^ΔCD4 mice. These mice developed an ALPS-like pathology characterized by splenomegaly and T cell infiltrates in various non-lymphoid tissues. No intestinal phenotype was observed, and serum levels of IgA were comparable to those of control mice (Salmena and Hakem, 2005). Similarly, mutations in the human *Casp8* gene have been identified in patients with clinical ALPS-like symptoms, although these patients also exhibited defective T cell activation (Chun et al, 2002). In contrast, patients

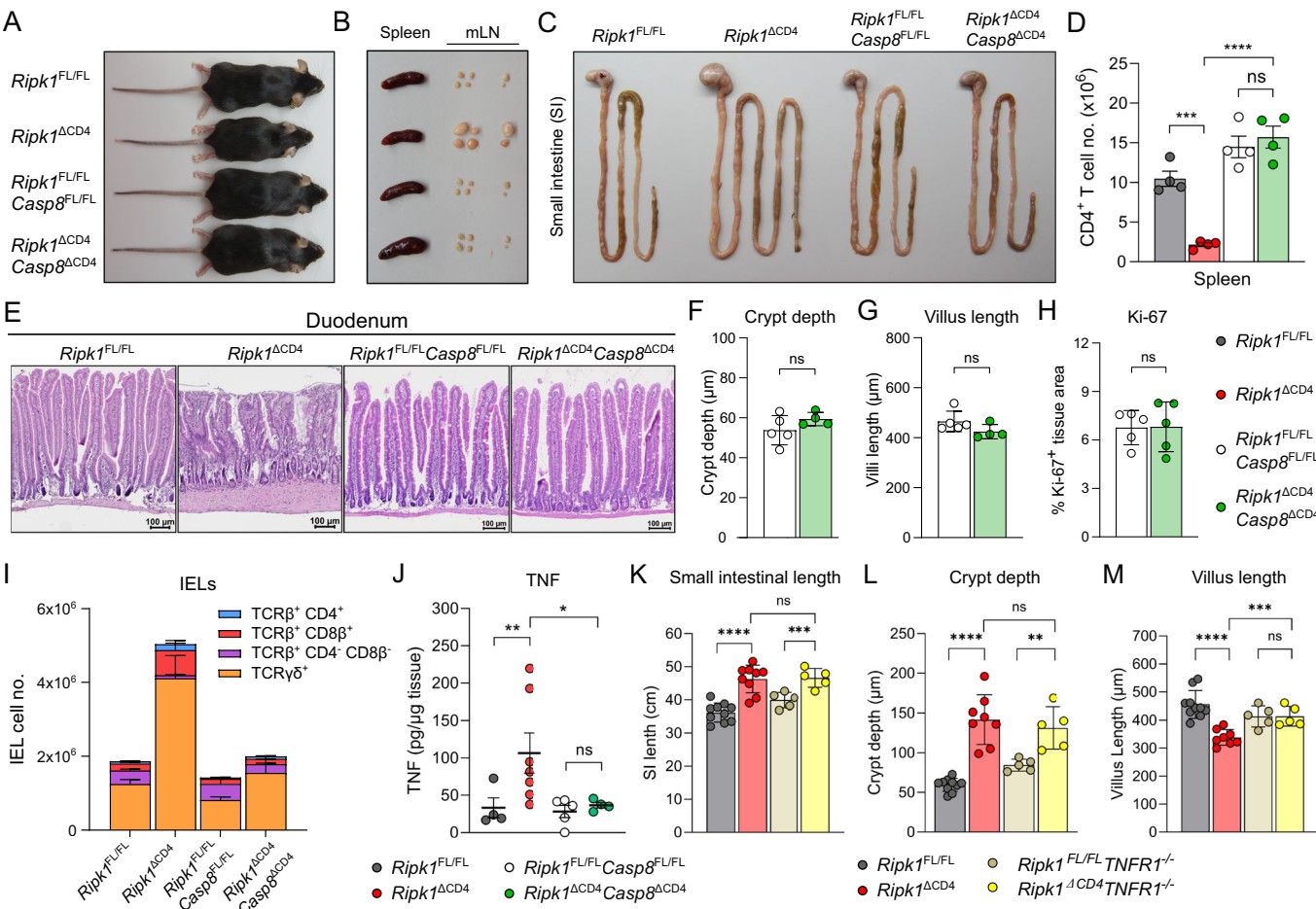

**Figure 7. Villus atrophy is TNFR1-dependent, and T cell ablation of caspase-8 prevents the development of intestinal pathology.**

(A–C) Representative images of (A) aged *Ripk1*ΔCD4 mice (n = 15) and *Ripk1*ΔCD4*Casp8*ΔCD4 mice (n = 16) alongside their respective littermate controls *Ripk1*FL/FL mice (n = 16) and *Ripk1*FL/FL*Casp8*FL/FL mice (n = 14), (B) their spleen and mLN as well as (C) their small intestines (SI). (D) Absolute numbers of CD4+ T cells in the spleen of *Ripk1*ΔCD4 and *Ripk1*ΔCD4*Casp8*ΔCD4 mice, and their respective *Ripk1*FL/FL and *Ripk1*FL/FL*Casp8*FL/FL littermates with n = 4 mice per group, measured by flow cytometry. (E) Representative H&E staining on duodenum sections from aged *Ripk1*ΔCD4 mice (n = 5) and *Ripk1*ΔCD4*Casp8*ΔCD4 mice (n = 4) and their respective littermate controls *Ripk1*FL/FL mice (n = 6) and *Ripk1*FL/FL*Casp8*FL/FL mice (n = 5). (F–H) Quantification of (F) the average crypt depth, (G) average villus length, and (H) Ki-67+ cells in the duodenum of *Ripk1*ΔCD4*Casp8*ΔCD4 mice (n = 4) and *Ripk1*FL/FL*Casp8*FL/FL littermates (n = 5). (I) Absolute numbers of TCRβ+CD4+, TCRβ+CD8β+, TCRβ+CD4-CD8β- and TCRγδ+ IELs in the SI epithelial layer of young *Ripk1*ΔCD4 mice (n = 4) and *Ripk1*ΔCD4*Casp8*ΔCD4 mice (n = 4) and their respective littermate controls *Ripk1*FL/FL mice (n = 4) and *Ripk1*FL/FL*Casp8*FL/FL mice (n = 3), measured by flow cytometry. The corresponding statistical analysis is depicted in Fig. EV5E. (J) Tissue concentrations of TNF in the SI of young *Ripk1*ΔCD4 mice (n = 7) and *Ripk1*ΔCD4*Casp8*ΔCD4 mice (n = 4) and their respective littermates *Ripk1*FL/FL (n = 4) and *Ripk1*FL/FL*Casp8*FL/FL (n = 5), measured by multiplex analysis (Meso Scale Discovery). (K–M) Quantification of the (K) absolute small intestine (SI) length, (L) Crypt depth, and (M) Villus length, of aged *Ripk1*ΔCD4 mice (n = 8) and *Ripk1*FL/FL littermates (n = 10), and *Ripk1*ΔCD4*Tnfr1*−/− mice (n = 5) and *Ripk1*FL/FL *Tnfr1*−/− littermates (n = 5). Data are representative of two (D, I–M) or three repeats (A–C, E–H). Data are shown as mean ± SEM, with means being represented by bars or horizontal lines and each dot representing an individual mouse. Statistical significance was calculated in Graphpad Prism by two-sided unpaired t-test with Welch's correction (F) p = 0.1821, (G) p = 0.123, (H) p = 0.9497, or two-way ANOVA on Log₂-transformed data (D) *Ripk1*FL/FL vs *Ripk1*ΔCD4: p = 0.0002, *Ripk1*ΔCD4 vs *Ripk1*ΔCD4*Casp8*ΔCD4: p < 0.0001, *Ripk1*FL/FL*Casp8*FL/FL vs *Ripk1*ΔCD4*Casp8*ΔCD4: p = 0.4399; (I) corresponding statistical analysis is depicted in Fig. EV5E; (J) *Ripk1*FL/FL vs *Ripk1*ΔCD4: p = 0.0034, *Ripk1*ΔCD4 vs *Ripk1*ΔCD4*Casp8*ΔCD4: p = 0.0182, *Ripk1*FL/FL*Casp8*FL/FL vs *Ripk1*ΔCD4*Casp8*ΔCD4: p = 0.8625; (K) *Ripk1*FL/FL vs *Ripk1*ΔCD4 (*TNFR1*+/+): p < 0.0001, *Ripk1*FL/FL vs *Ripk1*ΔCD4 (*TNFR1*−/−): p = 0.0045, *TNFR1*+/+ vs *TNFR1*−/− (*Ripk1*ΔCD4): p = 0.8363; (L) *Ripk1*FL/FL vs *Ripk1*ΔCD4 (*TNFR1*+/+): p < 0.0001, *Ripk1*FL/FL vs *Ripk1*ΔCD4 (*TNFR1*−/−): p = 0.0004, *TNFR1*+/+ vs *TNFR1*−/− (*Ripk1*ΔCD4): p = 0.3015; (M) *Ripk1*FL/FL vs *Ripk1*ΔCD4 (*TNFR1*+/+): p < 0.0001, *Ripk1*FL/FL vs *Ripk1*ΔCD4 (*TNFR1*−/−): p = 0.9755, *TNFR1*+/+ vs *TNFR1*−/− (*Ripk1*ΔCD4): p = 0.0009. ns = non-significant, *p < 0.05, **p < 0.01, ***p < 0.001, ****p < 0.0001. Young mice: 8–12 weeks old. Aged mice: >6 months old. IEL intraepithelial lymphocytes. Source data are available online for this figure.

with RIPK1 deficiency suffer from early-onset intestinal inflammation, immune deficiency, and arthritis (Uchiyama et al, 2019; Cuchet-Lourenço et al, 2018; Li et al, 2019). Mice lacking *Casp8* in T cells develop RIPK1-driven lymphoproliferative syndromes (Ch'en et al, 2011, 2008; Salmena and Hakem, 2005), whereas we demonstrate that the absence of *Ripk1* induces caspase-8-driven lymphopenia and intestinal malignancies. The absence of any

pathology in *Ripk1*ΔCD4*Casp8*ΔCD4 mice indicates that the pathologies induced by either *Ripk1* or *Casp8* ablation are both rescued.

Although patients with RIPK1 deficiency develop primary immunodeficiency associated with peripheral T cell lymphopenia (Sultan et al, 2022; Li et al, 2019), the impact of this deficiency on intestinal T cells remains unexplored. While human RIPK1 deficiency affects the entire body, the reported duodenal pathology

(Uchiyama et al, 2019) mirrors the one that is observed in *Ripk1*$^{\Delta CD4}$ mice with selective ablation of *Ripk1* in T cells. This parallel pathology between humans and mice indicates a central role of T cells in mediating intestinal pathology in humans with RIPK1 deficiency. Our findings indicates that patients with RIPK1 deficiency may benefit from bone marrow transplants to ameliorate systemic immune cell deficiencies.

In this study, we describe a mouse model that develops complex SI intestinal pathology, where the microbiome contributes to intestinal elongation, and TNF drives the villus atrophy. While the etiology of this genetic mouse model specifically targets the survival of αβT cells, it mimics the proximal SI pathology reported in human RIPK1 deficiency (Uchiyama et al, 2019). Additionally, this study reveals how intestinal T cell perturbation through RIPK1 ablation can result in specific pathological features observed in human enteropathies, such as celiac disease, autoimmune enteropathy and Crohn's disease (Vyhlidal et al, 2021; Lebwohl et al, 2018; Gentile et al, 2012). The similarity between these mice and the proximal SI manifestations observed in human patients may aid in unraveling novel downstream targets for human enteropathies, advancing clinical understanding in the field.

# Methods

### Reagents and tools table

| Antibodies | Source | Catalogue number (Cat#) |
|---|---|---|
| Anti-CD16/CD32 (clone 2.4G2) | BD Biosciences | 553142 |
| BUV395-conjugated anti-CD3 (145-2C11) | BD Biosciences | 563565 |
| BV605-conjugated anti-CD4 (RM4-5) | BD Biosciences | 563151 |
| BUV395-conjugated anti-CD25 (PC61) | BD Biosciences | 564022 |
| V450-conjugated anti-CD44 (IM7) | BD Biosciences | 560451 |
| BV421-conjugated anti-IL-10 (JES5-16E3) | BD Biosciences | 563276 |
| V450-conjugated anti-CD45 (30-F11) | BD Biosciences | 560501 |
| BUV395-conjugated anti-CD45 (30-F11) | BD Biosciences | 565967 |
| AF700-conjugated anti-CD45R/B220 | BD Biosciences | 557957 |
| PE-conjugated anti-TGF-β1 (TW7-16B4) | BD Biosciences | 141403 |
| PE-conjugated anti-CD62L (MEL-14) | BD Biosciences | 553151 |
| AF488-conjugated anti-FoxP3 (MF23) | BD Biosciences | 560407 |
| PE-conjugated anti-NKG2D (CX5) | BD Biosciences | 558403 |
| FITC-conjugated anti-TCRγδ | BD Biosciences | 553177 |
| PE-CF594-conjugated anti-TCRγδ | BD Biosciences | 563532 |
| AF488-conjugated anti-CD3 (17A2) | Biolegend | 100210 |
| FITC-conjugated anti-CD8β (YTS156.7.7) | Biolegend | 126606 |
| BV510-conjugated anti-CD8β (YTS156.7.7) | Biolegend | 126631 |
| PerCP-Cy5.5-conjugated anti-CD8β (YTS156.7.7) | Biolegend | 126610 |
| PE-Cy7-conjugated anti-CD8β (YTS156.7.7) | Biolegend | 126616 |

| Antibodies | Source | Catalogue number (Cat#) |
|---|---|---|
| PerCP-Cy5.5-conjugated anti-CD44 (IM7) | Biolegend | 103032 |
| APC-conjugated anti-TNF (MP6-XT22) | Biolegend | 554420 |
| BV785-conjugated anti-CD11b (M1/70) | Biolegend | 101243 |
| BV785-conjugated anti-CD19 (6D5) | Biolegend | 115543 |
| BV421-conjugated anti-CD45R/B220 (RA3-6B2) | Biolegend | 103239 |
| BV785-conjugated anti-F4/80 (BM8) | Biolegend | 123141 |
| BV785-conjugated anti-Ly-6G (1A8) | Biolegend | 127645 |
| BV711-conjugated anti-TCRβ (H57-597) | Biolegend | 563135 |
| BV785-conjugated anti-Ter-119 (TER-119) | Biolegend | 116245 |
| eFluor450-conjugated anti-CD8α (53-6.7) | eBioscience | 48-0081-82 |
| PE-Cy5-conjugated anti-CD11b (M1/70) | eBioscience | 11-0112-82 |
| AF700-conjugated anti-CD19 (1D3) | eBioscience | 56-0193 |
| PE-Cy7-conjugated anti-CD44 (IM7) | eBioscience | 25-0441-82 |
| PE-Cy7-conjugated anti-CD45 (30-F11) | eBioscience | 25-0451-82 |
| FITC-conjugated anti-CD62L (MEL-14) | eBioscience | 11-0621-82 |
| APC-conjugated anti-FoxP3 (FJK-16s) | eBioscience | 17-5773-82 |
| PE-Cy5-conjugated anti-Gr-1 (RB6-8C5) | eBioscience | 13-5931-82 |
| PE-Cy5-conjugated anti-Ter-119 (TER-119) | eBioscience | 13-5921-82 |
| Anti-RIPK1 | BD Biosciences | 610459 |
| Anti-caspase-8 | Abnova | MAB3429 |
| Anti-actin | MP Biomedicals | 69100 |
| HRP-linked anti-mouse IgG | GE Healthcare | NA931 |
| HRP-linked anti-rat IgG | GE Healthcare | NA935 |
| AF488-Labeled T22 tetramer | Tetramer Core Facility, NIH | 70603 |
| Fixable Viability Dye eFluor™ 780 | eBioscience | 65-0865-14 |
| Cell Stimulation Cocktail | Life Technologies | 00-4975-03 |
| BD Cytofix/Cytoperm™ | BD Biosciences | 554714 |
| Foxp3/Transcription Factor Staining Buffer Set | eBioscience | 00-5523-00 |

| Reagent | Source | Catalogue number |
|---|---|---|
| ACK lysis buffer | Lonza | 10548E |
| EDTA (2 mM) | Invitrogen | 15575020 |
| Dithioerythritol (DTE, 1 mM) | Sigma-Aldrich | D8255 |
| Percoll (37.5% solution) | Sigma-Aldrich | GE17-0891-01 |
| DNase I (40 μg/ml) | Roche | 10104159001 |
| Collagenase I (100 U/ml) | Gibco | 17100-017 |
| Zirconium oxide beads | Bertin | P000927-LYSK0-A |
| Food (for mice) | ssniff Spezialdiäten GmbH | V1534 |
| Ciprofloxacin | Sigma-Aldrich | 17850 |
| Ampicillin | Sigma-Aldrich | A9518 |
| Metronidazole | Sigma-Aldrich | M1547 |
| Vancomycin | Labconsult | DUC.V0155.0005 |
| Brain Heart Infusion Broth (BHI) | Sigma-Aldrich | 53286 |
| Brewer Thioglycolate Medium | Sigma-Aldrich | B2551 |
| U-PLEX Custom Biomarker Assay | Meso Scale Discovery | K15069M |

| Antibodies | Source | Catalogue number (Cat#) |
|---|---|---|
| Mouse Isotyping Panel 1 | Meso Scale Discovery | K15183B |
| **Mouse line** | **Reference and source** | **Strain number** |
| Cd4-Cre transgenic mice | Lee et al, 2001; purchased from Jackson Laboratory | 022071 |
| Ripk1$^{FL/FL}$ (Conditional Ripk1 mice) | Generated in-house at VIB Transgenic Core Facility (Takahashi et al, 2014) | Not applicable |
| T cell-specific RIPK1 knockout mice | Crossed Ripk1$^{FL/FL}$ with Cd4-Cre transgenic mice | Not applicable |
| RIPK1 kinase-dead knock-in mice (Ripk1K45A) | Berger et al, 2014; purchased from GlaxoSmithKline (GSK) | Not applicable |
| Casp8$^{FL/FL}$ (Conditional Casp8 mice) | Beisner et al, 2005; purchased from Jackson Laboratory | 027002 |
| T cell-specific Ripk1 and Casp8 double knockout mice | Crossed Casp8$^{FL/FL}$ with Ripk1FL/+ Cd4-CreTg/+ mice | Not applicable |
| Tnfr1-deficient mice (Tnfr1$^{-/-}$) | Pfeffer et al, 1993; purchased from Jackson Laboratory | 002818 |
| Full-body Tnfr1$^{-/-}$ and T cell-specific Ripk1-deficient mice | Crossed Tnfr1$^{-/-}$ with Ripk1FL/+ Cd4-Cre$^{Tg/+}$ | Not applicable |
| CD45.1 | Purchased from Jackson Laboratory | 002014 |
| CD45.1/2 | Product of mating CD45.1 with CD45.2 (F1 generation) | Not applicable |
| **Sequence of primers** | **Alleles identified by primers** | **Additional information** |
| 5'-GGCAAACACCTTTAATCCAAGCCTGGTC-3' | Wild type and loxP-site flanked Ripk1 alleles - Forward | Wild-type: 287 bp, Floxed: 366 bp |
| 5'-CCATGGCTGCAAACACCTAAACCTGAAG-3' | Wild type and loxP-site flanked Ripk1 alleles - Reverse | Wild-type: 287 bp, Floxed: 366 bp |
| 5'-GCCTGCATTACCGGTCGATGCAACGA-3' | Cd4-Cre construct - Forward | 800 bp DNA fragment when the transgene is present |
| 5'-GTGGCAGATGGCGCGGCAACACCAT-3' | Cd4-Cre construct - Reverse | 800 bp DNA fragment when the transgene is present |
| 5'-CTCTGATTGCTTTATAGGACACAGCA-3' | Ripk1K45A mice (knock-in) - Forward | Wild-type: 575 bp, Knock-in: 473 bp |
| 5'-GTCTTCAGTGATGTCTTCCTCGTA-3' | Ripk1K45A mice (knock-in) - Reverse | Wild-type: 575 bp, Knock-in: 473 bp |
| 5'-TTGAGAACAAGACCTGGGGACTG-3' | loxP-site flanked Casp8 alleles - Forward | Wild-type: 200 bp, Floxed: 300 bp |
| 5'-GGATGTCCAGGAAAAGATTTGTGTC-3' | loxP-site flanked Casp8 alleles - Reverse | Wild-type: 200 bp, Floxed: 300 bp |
| 5'-CTGGAAGTGTGTCTCAC-3' | Tnfr1 alleles - Forward | Wild-type: 1400 bp, Knockout: 950 bp |
| 5'-CCAAGCGAAACATCGCATCGAGCGA-3' | Tnfr1 alleles - Reverse | Wild-type: 1400 bp, Knockout: 950 bp |
| **Software/Program** | **Source** | **Identifier/Version** |
| Zeiss Zen blue software | Carl Zeiss | Version 3.1 |
| The R Project for Statistical Computing | www.R-project.org, R Core Team (2021) | Version 4.1.1 |
| QuPath | Open Source, Bankhead et al (2017) | Version 0.2.3 |
| FACSDiva™ software | BD Biosciences | Version 9.0 |
| FlowJo™ | Flowjo.com | Version 10 |
| Cell Ranger | 10x Genomics | Version 5.0.0 |
| Seurat | Open Source (R package), Hao et al (2021) | Version 4.0.2 |
| pheatmap | Open Source (R package), Kolde et al (2019) | Version 1.0.12 |
| GraphPad Prism | GraphPad Software | Version 10.3.0 |
| ggplot2 | Open Source (R package), Wickham et al (2016) | Version 3.5.1 |
| pathfindR | Open Source (R package), Ulgen et al (2019) | Version 2.4.1 |

## Mice

All conditional KO mice were on a C57BL/6J background, and both male and female sexes were used for all experiments. The different mouse stains can be found in the Reagents and Tools Table and above: Conditional Ripk1 mice (Ripk1$^{FL/FL}$) were generated in-house at VIB Transgenic Core Facility (TCF), as previously described (Takahashi et al, 2014). Cd4-Cre transgenic mice (Lee et al, 2001) were purchased from Jackson Laboratory (Strain #:022071). T cell-specific RIPK1 knockout mice were created by crossing Ripk1$^{FL/FL}$ mice with Cd4-Cre transgenic mice. Mice were bred using one heterozygous parent (Ripk1$^{FL/+}$ Cd4-Cre$^{Tg/+}$) and one Ripk1$^{FL/FL}$ parent because the use of homozygous parents (Ripk1$^{FL/FL}$ Cd4-Cre$^{Tg/+}$) resulted in reduced litter sizes. Ripk1$^{FL/+}$ Cd4-Cre$^{+/+}$ and Ripk1$^{FL/FL}$ Cd4-Cre$^{+/+}$ mice were used as control genotypes. RIPK1 kinase-dead knock-in mice (Ripk1$^{K45A}$) (Berger et al 2014), were purchased from GlaxoSmithKline (GSK). Conditional Casp8 mice (Casp8$^{FL/FL}$) (Beisner et al, 2005) were purchased from Jackson Laboratory (Strain #:027002). T cell-specific Ripk1 and Casp8 double knockout mice were created by crossing Casp8$^{FL/FL}$ mice with Ripk1$^{FL/+}$ Cd4-Cre$^{Tg/+}$ mice, allowing for the study of combined gene knockouts in T cells. Tnfr1-deficient mice (Tnfr1$^{-/-}$) (Pfeffer et al, 1993), were purchased from Jackson Laboratory (Strain #:002818). Full-body Tnfr1$^{-/-}$ and T cell-specific Ripk1-deficient mice were created by crossing Tnfr1$^{-/-}$ mice with Ripk1$^{FL/+}$ Cd4-Cre$^{Tg/+}$ mice. Experimental mouse lines were cohoused, and littermate controls were used in all experiments to ensure appropriate comparisons. CD45.1 were from the Jackson Laboratory (Strain #: 002014), and CD45.1/2 mice were the product of mating CD45.1 with CD45.2 (F1 generation). They were bred and housed in individually ventilated cages at the VIB Center for Inflammation Research under specific pathogen-free (SPF) conditions in a temperature-controlled (21 °C) animal facility with a 14/10-h light/dark cycle. Water and food (ssniff Spezialdiäten GmbH, V1534) were provided ad libitum. All experiments on mice were conducted according to institutional, national, and European animal regulations. All animal protocols were approved by the ethics committee of Ghent University. Mice defined in the text as 'Young' were between 8 to 12 weeks, and mice defined as 'Aged' were at least 6 months old. For assessment of body weight and survival, mice were monitored and weighed weekly or biweekly. For depletion of the intestinal microbiome, breeding couples were treated with an antibiotic cocktail consisting of 200 mg ciprofloxacin (Sigma-Aldrich, 17850), 1 g ampicillin (Sigma-Aldrich, A9518), 1 g metronidazole (Sigma-Aldrich, M1547) and 500 mg vancomycin (Labconsult, DUC.V0155.0005) per litre of drinking water. Treatment began 2 weeks before the expected birth of their offspring. After birth, treatment was continued on the offspring for 28 weeks to ensure sufficient time for the intestinal pathology to occur. Drinking water with freshly dissolved antibiotics was changed every 2–3 days. After the antibiotic-treated mice were sacrificed, tissues were harvested, and the presence of intestinal microflora was determined by culturing fecal samples in liquid cultures of brain heart infusion broth (Sigma-Aldrich, 53286) and Brewer thioglycolate medium (Sigma-Aldrich, B2551) for 2 days at 37 °C. No significant bacterial growth was observed in the liquid cultures from antibiotic-treated mice, indicating effective antibiotic treatment. All experiments on mice were conducted in accordance

with the Declaration of Helsinki and approved by the IRC-UGent ethical committee under the reference numbers (EC2019-009, EC2018-088, and EC2020-045).

## Genotyping

Genotyping of the different mouse lines was performed using conventional PCR and the primers can be found in the Reagents and Tools Table describing the specific primer sequences and the size of the resulting DNA fragments.

## Histology

Mice were euthanized by cervical dislocation at indicated time points. Small intestines and colons were isolated, and the small intestines were divided into duodenum, jejunum and ileum. The three intestinal fragments and colons were flushed with 1x PBS at 4 °C, either rolled up into Swiss rolls and fixed overnight in 4% formaldehyde at 4 °C. After fixation, tissues were dehydrated using the Shandon Citadel tissue processor (Thermo Scientific), embedded in paraffin using standardized methods and cut at 4 µm thickness. Sections were stained with hematoxylin and eosin (H&E) using the Thermo Shandon Varistain V24-4 (Thermo Scientific). Terminal deoxynucleotidyl transferase dUTP nick end labeling (TUNEL) assay was combined with a CD45 immunofluorescent staining. Sections were deparaffinized, rehydrated and antigen retrieval was performed using an antigen unmasking solution (Vector Laboratories, VEC.H-3300) in a Pick cell cooking unit. Next, TUNEL assay (In situ cell death detection kit TMR red) was performed following the manufacturer's instructions (Roche, 12156792910), followed by a blocking step with 5% goat serum in PBS for 30 min and an overnight incubation with a rabbit anti-CD45 primary antibody (1/400, Abcam, ab10558). Additionally, sections were incubated with a biotinylated secondary antibody (Vector Laboratories, BA-1000) followed by streptavidin-Dylight633 (ThermoFisher, 21844) and DAPI (invitrogen, D21490). For Immunohistochemistry, sections were incubated overnight with a rabbit anti-Ki67 antibody (1/1000, Cell Signaling, 12202S) followed by a biotinylated secondary antibody (Vector Laboratories, BA-1000). An avidin-biotin system (ABC, Vector Laboratories, PK4000) was added and peroxidase activity was detected using 3,5-di-amino-benzidine (DAB, Vector Laboratories, SK-4105). Images were acquired using a Zeiss Axioscan Z.1 slide scanner (Carl Zeiss, Jenna, Germany) at 20x magnification. Quantification of villus length and crypt depth was performed using Zeiss Zen 3.1 blue software, and researchers were always blinded to the individual groups. Quantification of Ki-67, TUNEL, and CD45 stainings was performed using scripts provided by the VIB Bioimaging Core (Ghent, Belgium) ran on QuPath-0.2.3 software. For each sample, three separate regions per Swiss roll were quantified, and the average was calculated.

## Scanning electron microscopy (SEM)

Cross-sections were cut from the duodenum and immediately incubated in freshly prepared fixative containing 2% paraformaldehyde (EMS), 2.5% gluteraldehyde (EMS) in 0.1 M Sodium Cacodylate (EMS) buffer, pH 7.4) overnight at 4 °C. The fixative was removed by washing the samples in 0.1 M cacodylate buffer, and samples were then incubated in 2% osmium ($OsO_4$, EMS) in 0.1 M cacodylate buffer for 30 min at RT. After washing in $H_2O$, the samples were dehydrated using solutions of increasing ethanol concentration (50%, 70%, 85%, 95%, 2x 100%), followed by acetone. The samples were subsequently dried in a critical point dryer (Leica EM CPD300) and mounted on an aluminum stub (EMS) using carbon adhesive tape (EMS). Samples were coated with 5 nm of platinum (Quorum Q150T ES), and SEM imaging was performed using a Zeiss Crossbeam 540.

## Generation of mixed bone marrow chimeras

CD45.1 recipient mice received a sublethal irradiation dose (800 cGy) to prepare them for bone marrow reconstitution. At least 4 h later, the recipient mice were injected intravenously with a 1:1 mixture of BM cells ($4 \times 10^6$ cells in total) derived from CD45.1/2 and either $Ripk1^{FL/FL}$ or $Ripk1^{\Delta CD4}$ (both CD45.2) donors. Irradiated mice were randomly assigned to the individual bone marrow chimera groups. Validation of the reconstitution was performed 8 weeks later by collection of tail vein blood. Osmotic lysis of red blood cells was carried out using ACK lysis buffer (10548E, Lonza, Basel, Switzerland), and lymphocytes were analyzed by flow cytometry. Twelve weeks after reconstitution of the bone marrow, the mice were euthanized, and ratios of CD45.1/2 and CD45.2T cell numbers in IEL and LP were analyzed by flow cytometry.

## Preparation of single cells for flow cytometry

Thymus, spleens, lymph nodes and Peyer's patches were harvested and stored in ice-cold PBS containing 3% heat-inactivated fetal calf serum (FCS) and 2 mM EDTA (Invitrogen, 15575020). Tissues were smashed on top of a 70 µm cell strainer using the plunger of a 3-ml syringe, and single cells were subsequently collected by centrifugation at $500 \times g$ and 4 °C for 5 min. For the purification of splenocytes, an additional osmotic lysis of red blood cells was performed using ACK lysis buffer (Lonza, 10548E). Small intestines were dissected and flushed with RPMI containing 5% heat-inactivated FCS to remove feces and mucus. Peyer's patches were carefully removed, and the intestines were dissected longitudinally and transversely into 0.5 cm pieces. For isolation of immune cells within the epithelial layer, the fragments were incubated in Hanks' Balanced Salt Solution (HBSS) supplemented with 10 mM HEPES buffer, 25 mM $NaHCO_3$, 10% FCS and 1 mM dithioerythritol (DTE, Sigma-Aldrich, D8255) for 20 min on a shaker at 37 °C. Intra-epithelial lymphocytes (IEL) were further enriched by resuspension in a 37.5% Percoll solution followed by gradient centrifugation (Sigma-Aldrich, GE17-0891-01), a technique designed to separate lymphocytes from other cell types based on their density, such as epithelial cells. The remaining tissue pieces were further incubated in HBSS containing 5 mM HEPES and 1.3 mM EDTA for 20 min on a shaker at 37 °C to remove the remaining epithelial cells. The tissue pieces were then digested in RPMI supplemented with 10% FCS, 1 mM $MgCl_2$, 1 mM $CaCl_2$, 40 µg/ml DNAse I (Roche, 10104159001), and 100 U/ml collagenase I (Gibco, 17100-017) for 45 min at 37 °C on a shaker. This enzymatic digestion helped to break down the extracellular matrix, facilitating further cell isolation. Afterwards, LP lymphocytes were enriched by gradient centrifugation following resuspension of cell suspension in a 37.5% Percoll solution.

## Flow cytometry

Single-cell suspensions were obtained as described in the previous section, and the antibodies can be found in Reagents and Tools Table. Staining for cell viability was done using Fixable Viability Dye eFluor™ 780 (eBioscience). Intracellular stainings for FoxP3 were performed using the Foxp3/Transcription Factor Staining Buffer Set (eBioscience, 00-5523-00). For intracellular cytokine measurements, cells were first extracted from each tissue as mentioned above and incubated with Cell stimulation cocktail (Cat# 00-4975-03, Life Technologies) for 4 h. For the staining procedure, the cells were first stained extracellularly, then fixed and permeabilized using BD Cytofix/Cytoperm™ (Cat# 554714, BD Biosciences), followed by intracellular staining. Flow cytometry was performed on an LSRFortessa™ 5-laser or FACSymphony™ cytometer (BD Biosciences) using FACSDiva™ software, and data were analyzed using FlowJo™ v10 software.

## Measurement of cytokine concentrations in small intestine

Intestinal tissue pieces were homogenized with zirconium oxide beads (Bertin, P000927-LYSK0-A) using the Precellys 24 tissue homogenizer (Bertin). Cytokine concentrations, including IFN-γ, IL-4, IL-5, IL-6, IL-9, IL-10, IL-13, IL-17A, IL-22, TNF, and TGF-β were measured using a U-PLEX Custom Biomarker assay (Meso Scale Discovery, K15069M) according to the manufacturer's instructions. Measurements were conducted on the Meso Quickplex SQ 120 (Meso Scale Discovery) and normalized to tissue weight.

## Measurement of immunoglobulins in serum

Mice blood samples were collected and centrifuged at $1500 \times g$ for 10 min to separate serum. IgA levels were quantified using the Mouse Isotyping Panel 1 assay (Meso Scale Discovery, K15183B) following the manufacturer's instructions. Measurements were performed on the Meso Quickplex SQ 120 (Meso Scale Discovery).

## Western blotting

FACS-sorted γδT IELs (Gating: live, CD3$^+$, TCRγδ$^+$, TCRβ$^-$) were collected and lysed in 1x Laemmli buffer containing 50 mM Tris-HCl pH 6.8, 2% sodium dodecyl sulfate (SDS), and 10% glycerol. Lysates were boiled for 10 min at 95 °C, and proteins were separated by SDS-PAGE before being transferred to a nitrocellulose membrane. Immunoblotting was performed by incubating at 4 °C overnight using the following primary antibodies: anti-RIPK1 (#610459, 1:2000, BD Biosciences), anti-caspase-8 (MAB3429, 1:1000, Abnova, Taipei, Taiwan) and anti-actin (69100, 1:20,000, MP Biomedicals, Irvine, CA, USA) followed by a 1 h incubation at RT using the following secondary antibodies: HRP-linked anti-mouse IgG (NA931, 1:3000, GE Healthcare, Chicago, IL, USA) and HRP-linked anti-rat IgG (NA935, 1:3000, GE Healthcare). FACS sorting was performed on a FACSAria™ III cell sorter (BD Biosciences).

## Sorting cells for scRNA-seq

Single cells were prepared from the small intestine as previously described. The cells were subsequently stained with DAPI for cell viability, and TruStain FcX Block (BioLegend, cat 101320). Antibody staining comprised fluorochrome-conjugated monoclonal antibodies recognizing CD45 (eBioscience, clone 30-F11), the mouse cell surface protein antibody panel containing 165 oligo-conjugated antibodies, 9 TotalSeq-A isotype controls (TotalSeq-A, BioLegend) and the unique TotalSeq-A cell hashing antibody (BioLegend). Approximately, 70,000 CD45$^+$ cells and 30,000 CD45$^-$ cells were sorted per sample using FACSAria™ II and FACSAria™ III cell sorters (BD Biosciences) and afterwards combined for sequencing.

## Single-cell RNA sequencing (scRNA-seq)

Three biological replicates per genotype were multiplexed using TotalSeq-A cell hashing antibodies (Biolegend) to facilitate efficient sample tracking. Sorted IEL single-cell suspensions were diluted to an approximate concentration of 1000 cells/µl and processed using a Chromium GemCode Single Cell Instrument (10x Genomics) to generate single-cell gel beads-in-emulsion (GEM). The scRNA-Seq libraries were prepared using the GemCode Single Cell 3′ Gel Bead and Library kit, version 3.1 (10x Genomics) following the manufacturer's instructions. Amplification primers (3 nM each, 5′ CCTTGGCACCCGAGAATT*C*C - 5′GTGACTGGAGTTCAG ACGTGTGC*T*C) were added for cDNA amplification to enrich the TotalSeq-A cell surface and hashing protein oligos respectively. Size selection with SPRIselect Reagent Kit (Beckman Coulter, B23318) was used to separate amplified cDNA molecules for 3′ gene expression and cell surface protein construction. TotalSeq-A protein library construction, including sample index PCR using Illumina's Truseq Small RNA primer sets and SPRIselect size selection, was carried out following the manufacturer's instructions. The cDNA content of pre-fragmentation and post-sample index PCR samples was analyzed using the 2100 BioAnalyzer (Agilent) to ensure consistency and quality control. Sequencing libraries were loaded onto an Illumina NovaSeq flow cell at the VIB Nucleomics core with sequencing settings aligned with the recommendations of 10x Genomics. The libraries were pooled in a 70:20:10 ratio for the 3′ gene expression, cell surface, and hashing protein samples, respectively. The Cell Ranger pipeline (10x Genomics, version 5.0.0) was used to demultiplex samples and generate FASTQ files for read 1, read 2, and the i7 sample index for the gene expression, cell surface, and hashing protein libraries. Read 2 of the gene expression libraries was mapped to the mouse reference genome (GRCm38.99) to identify and annotate transcriptomic features. Barcode processing, filtering of unique molecular identifiers, and gene counting were performed using the Cell Ranger suite. CITE-seq reads were quantified using the feature-barcoding functionality to associate protein expression with the transcriptomic data. The mean reads per cell across all gene expression libraries was 24,100, with a sequencing saturation of 55%, as calculated by Cell Ranger. In total, eight individual single-cell libraries were created in this study, containing 86,180 cells.

## scRNA-seq data analysis

Separate samples were merged and the aggregate was processed using the Seurat (v 4.0.2) pipeline with default parameters unless specified otherwise. The following functions were used sequentially: NormalizeData, FindVariableFeatures, ScaleData, RunPCA (npcs = 150),

FindNeighbours (dims = 1:50), FindClusters (resolution = 1.3), RunTSNE (dims = 1:50), and RunUMAP (dims = 1:50). Clusters were further curated manually based on marker gene expression to ensure accurate identification and grouping of cell populations. Differentially expressed genes were calculated using the FindMarkers function with an adjusted *p* value of <0.05. Dot plots and UMAP plots were created using the DotPlot and DimPlot functions, respectively. Heatmaps were made using an in-house built function based on the pheatmap package (v 1.0.12). Volcano plots were generated in Graphpad Prism. KEGG analysis was performed using pathfindR (v 2.4.1) and visualized with ggplot2 (v 3.5.1) (Ulgen et al, 2019).

## Statistical analysis and study design

Effect size and sample size calculation were performed using G*power 3.1.9.7. software with an α-error of 0.05 and a power of 80%. Flow samples were only rejected in case high levels of death cells were determined following acquisition (making them unrepresentable) or if the genotype of the mouse was uncertain. The number of mice (*n*) for each experiment is reported in the figure legends. The test used to evaluate differences between condition groups is specified in the figure legend. Unpaired T-test were conducted as two-sided tests with Welch's correction. For comparisons involving more than two experimental groups, differences in variances between treatment groups were tested with an F-test. If the F-test indicated different variances between conditions, statistical analysis were conducted on Log2-transformed values, as specified in the figure legend. Due to the type of the data, normal distribution was assumed when appropriate but not formally tested. In case the normal distribution could not be assumed or the variances remained unequal after Log2 transformation, a non-parametric test was applied as indicated in figure legends. Statistical analyses were carried out using GraphPad Prism version 10.3.0 (Graphpad Software Inc., La Jolla, CA).

## Data availability

The single-cell sequencing data has been deposited in the Gene Expression Omnibus under the ID: GSE263990.

The source data of this paper are collected in the following database record: biostudies:S-SCDT-10_1038-S44319-025-00441-5.

## Peer review information

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

## Acknowledgements

We thank the VIB Single Cell Core, VIB imaging core, and VIB Flow Core Ghent, for support and access to the instrument park (vib.be/core-facilities). Additionally, we thank the VIB-UGent animal house staff. We thank Hilde Cheroutre for the valuable discussion. Some figures were created in BioRender: (https://BioRender.com/d45y106 and https://BioRender.com/d36q900). All authors have read or provided comments on the manuscript. We thank CRIG, GGIG consortia, and VIB for their support. JH held an FWO PhD fellowship (1S44919N). PV is a senior full professor at Ghent University and senior PI at the VIB-UGent Center for Inflammation Research (IRC). PT was funded through an FWO (12U8318N) and BOF-UGent postdoc fellowships (BOF20/PDO/027). Research in the Vandenabeele unit is supported by the FWO (research grants G.0E04.16N, G.0C76.18N, G.0B71.18N, G.0B96.20N, G.0A93.22N, EOS MODEL-IDI Grant (30826052), and EOS CD-INFLADIS (40007512)), grants from the Special Research Fund UGent (Methusalem grant BOF16/MET_V/007, BOF22/MET_V/007, iBOF ATLANTIS grant 20/IBF/039), grants from the Belgium Foundation against Cancer (UICC: F/2016/865, F/2020/1505).

## Author contributions

**Jelle Huysentruyt**: Conceptualization; Formal analysis; Investigation; Methodology; Writing—original draft. **Wolf Steels**: Formal analysis; Investigation; Methodology. **Mario Ruiz Perez**: Formal analysis; Investigation; Methodology. **Bruno Verstraeten**: Data curation; Software; Formal analysis; Methodology. **Tatyana Divert**: Investigation; Methodology. **Kayleigh Flies**: Methodology. **Kelly Lemeire**: Methodology. **Nozomi Takahashi**: Conceptualization; Supervision; Funding acquisition; Investigation; Project administration; Writing—review and editing. **Elke De Bruyn**: Formal analysis; Investigation; Methodology. **Marie Joossens**: Conceptualization; Data curation; Formal analysis; Supervision; Investigation. **Andrew S Brown**: Formal analysis; Methodology. **Bart N Lambrecht**: Resources; Supervision; Methodology. **Wim Declercq**: Conceptualization; Resources; Supervision; Writing—review and editing. **Tom Vanden Berghe**: Conceptualization; Supervision; Investigation; Writing—review and editing. **Jonathan Maelfait**: Conceptualization; Methodology; Writing—review and editing. **Peter Vandenabeele**: Conceptualization; Resources; Supervision; Funding acquisition; Investigation; Writing—original draft; Project administration; Writing—review and editing. **Peter Tougaard**: Conceptualization; Resources; Formal analysis; Supervision; Investigation; Methodology; Writing—original draft; Writing—review and editing.

Source data underlying figure panels in this paper may have individual authorship assigned. Where available, figure panel/source data authorship is listed in the following database record: biostudies:S-SCDT-10_1038-S44319-025-00441-5.

## Disclosure and competing interests statement

The authors declare no competing interests.

# Expanded View Figures

**Figure EV1. Selective deletion of *Ripk1* in conventional T cells results in small intestinal inflammation and elongation.**

(A) Distribution of pups with indicated genotypes from crosses between *Ripk1*^FL/FL^; *Cd4-Cre*^+/+^ and *Ripk1*^Fl/+^; *Cd4-Cre*^Tg/+^ parents. Bars represent the observed percentage of mice with each genotype within a total population of $n = 205$ mice. The expected percentage is denoted by the horizontal dotted line, and the *p*-value resulting from a chi-square test are indicated. (B) Absolute numbers of CD4^+^ and CD8^+^ T cells in the mesenteric lymph nodes (mLN) of *Ripk1*^ΔCD4^ mice and *Ripk1*^FL/FL^ littermates, measured by flow cytometry. (C) Representative image of the full colon and caecum of an aged *Ripk1*^ΔCD4^ mouse and *Ripk1*^FL/FL^ littermate. (D) Quantification of the absolute length of the colon of young *Ripk1*^ΔCD4^ mice ($n = 22$) and *Ripk1*^FL/FL^ littermates ($n = 22$) and aged *Ripk1*^ΔCD4^ mice ($n = 25$) and *Ripk1*^FL/FL^ littermates ($n = 22$). (E) Representative images of an aged homozygous *Ripk1*^ΔCD4^ mouse, heterozygous *Ripk1*^Fl/+^;*Cd4-Cre*^Tg/+^ mouse and *Ripk1*^FL/FL^ littermate with respective spleen, mLN and small intestine (SI). (F) Representative images of an aged *Ripk1*^K45A^ mouse and *Ripk1*^+/+^ littermate with respective spleen, mLN and small intestine (SI). (G) Quantification of the absolute SI length of young *Ripk1*^K45A^ mice ($n = 4$) and *Ripk1*^+/+^ littermates ($n = 3$) and aged *Ripk1*^K45A^ mice ($n = 7$) and *Ripk1*^+/+^ littermates ($n = 6$). (H) Tissue concentrations of IL-4, IL-5, IL-6, IL-10 IL-13, and TGF-β in the SI of young *Ripk1*^ΔCD4^ mice ($n = 13$) and *Ripk1*^FL/FL^ littermates ($n = 11$), and aged *Ripk1*^ΔCD4^ mice ($n = 17$) and *Ripk1*^FL/FL^ littermates ($n = 16$), were measured using multiplex assays (Meso Scale Discovery). Data are shown as mean ± SEM, with means represented by bars or horizontal lines and each dot representing an individual mouse. Data are representative of at least three independent repeats (**C**, **F**), or are combined from two independent repeats (**B**). Statistical significance was calculated in Graphpad Prism by (**A**) chi-square test, (**B**) Fisher's LSD two-way ANOVA on Log$_2$-transformed data CD4^+^: $p = 0.0006$, CD8β^+^: $p < 0.0001$, (**D**, **G**) Fisher's LSD two-way ANOVA on absolute values, or (**H**) two-sided unpaired T test with Welch's correction. ns non-significant, na not applicable, ***$p < 0.001$, ****$p < 0.0001$. Young mice: 8–12 weeks old. Aged mice: >6 months old. Source data are available online for this figure.

 

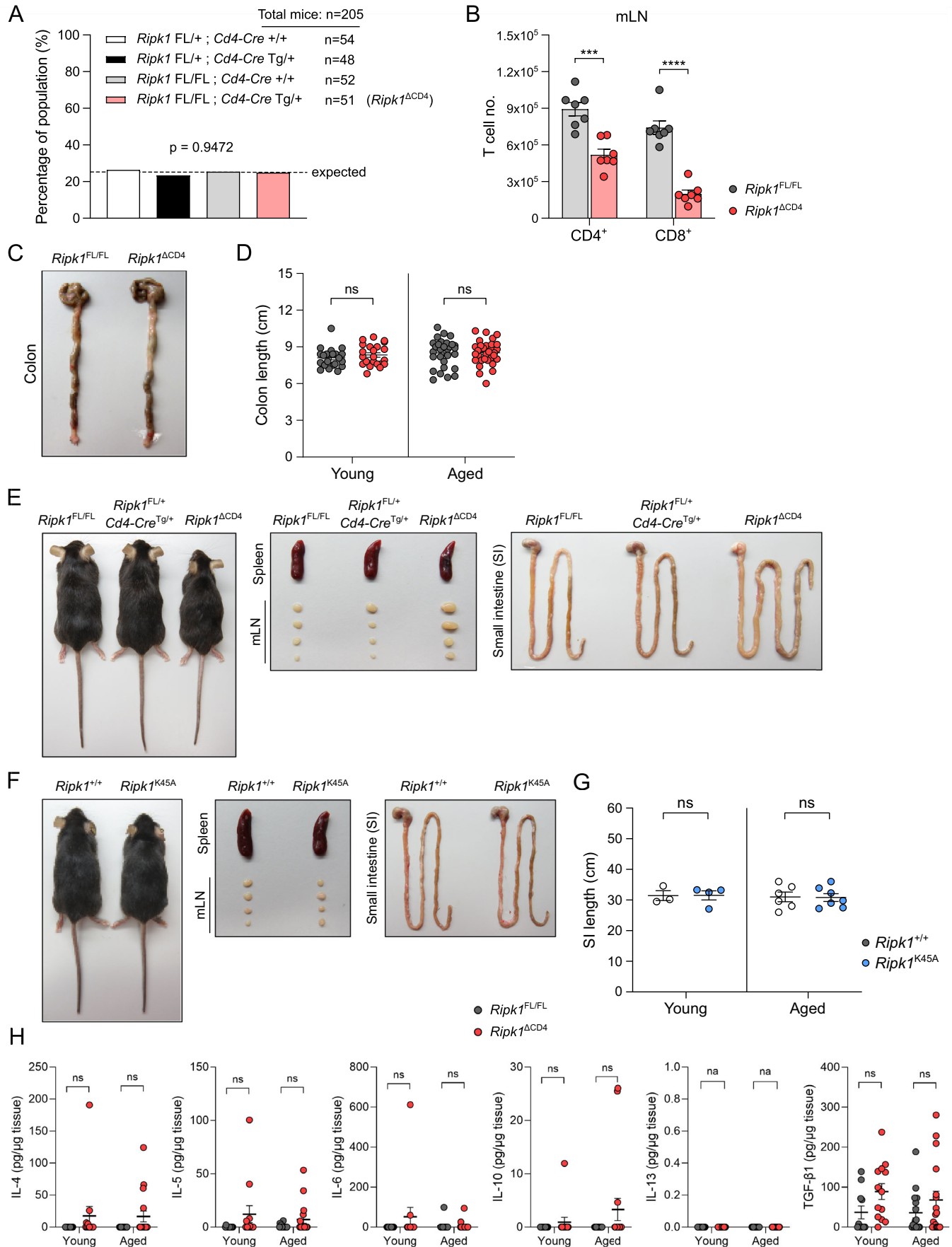

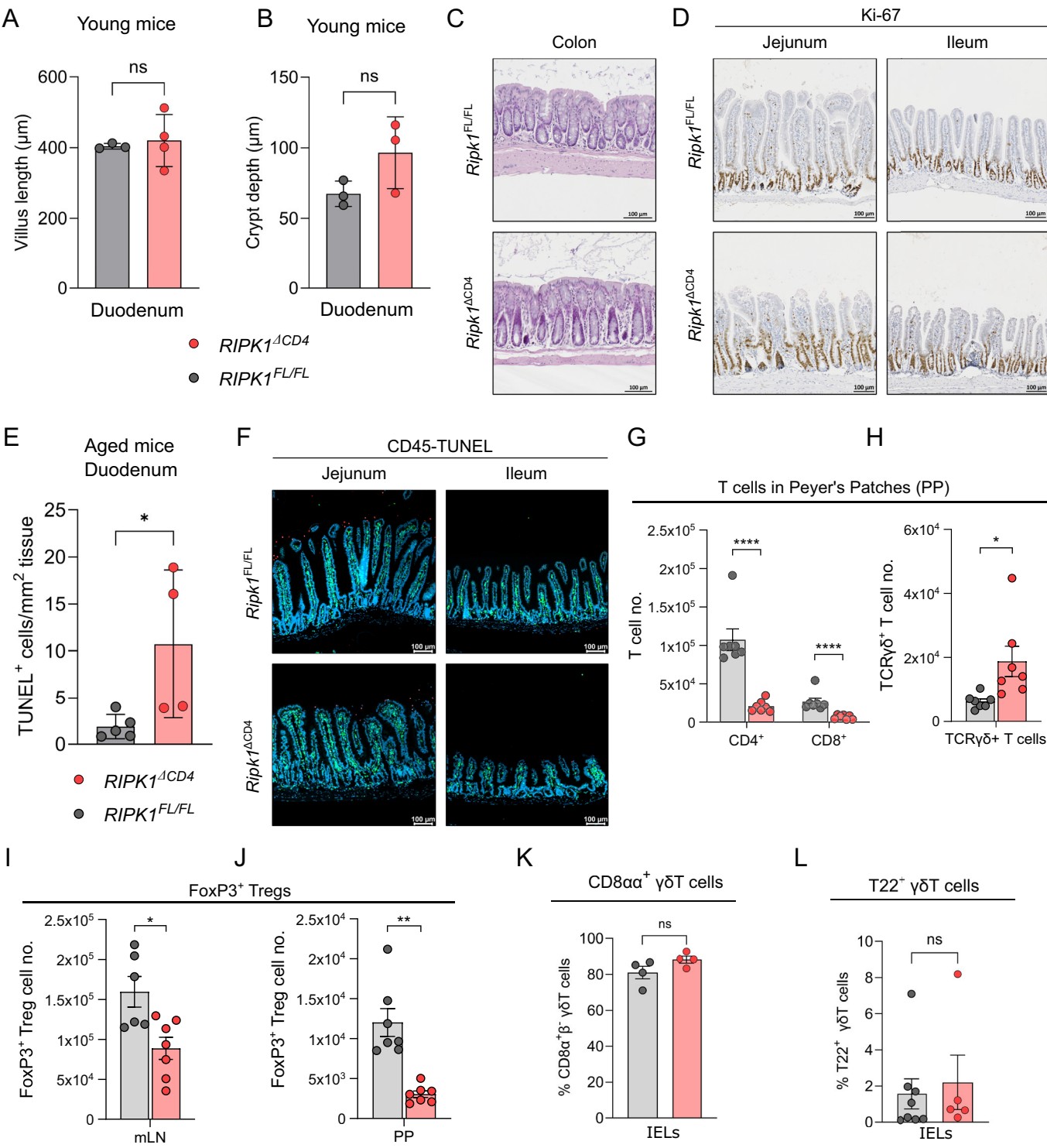

**Figure EV2.** **The duodenum of *Ripk1*^ΔCD4 mice displays villus atrophy, crypt hyperplasia, and immune cell infiltration.**

(A, B) Quantification of (A) the average villus length and (B) the average crypt depth in the duodenum of young *Ripk1*^ΔCD4 and *Ripk1*^FL/FL littermates (*n* = 3). (C) H&E staining on sections of the colon of an aged *Ripk1*^ΔCD4 mouse and *Ripk1*^FL/FL littermate. (D) Ki-67 staining on sections of the jejunum and ileum of aged *Ripk1*^ΔCD4 mice and *Ripk1*^FL/FL littermates. (E) Quantification of TUNEL staining in the duodenum of aged *Ripk1*^ΔCD4 mice (*n* = 4) and *Ripk1*^FL/FL littermates (*n* = 5), representing TUNEL-positive cells/mm² tissue area. (F) CD45-TUNEL staining on sections of the jejunum and ileum of aged *Ripk1*^ΔCD4 mice (*n* = 4) and *Ripk1*^FL/FL littermates (*n* = 5). (C, D, F) Images are representative of *Ripk1*^ΔCD4 mice (*n* = 4) and *Ripk1*^FL/FL littermates (*n* = 5). (G) Absolute numbers of CD4^+ and CD8^+ T cells in the Peyer's patches (PP) of young *Ripk1*^ΔCD4 mice and *Ripk1*^FL/FL littermates, measured by flow cytometry. (H) Absolute numbers of TCRγδ^+ T cells in the PP of young *Ripk1*^ΔCD4 mice and *Ripk1*^FL/FL littermates, measured by flow cytometry. Data are obtained from *n* = 7 mice per group (G, H). (I, J) Absolute numbers of FoxP3^+ Tregs in the mLN (I) and PP (J) of young *Ripk1*^ΔCD4 mice and *Ripk1*^FL/FL littermates, measured by flow cytometry. (I, J) Data are obtained from *Ripk1*^ΔCD4 mice (*n* = 6) and *Ripk1*^FL/FL littermates (*n* = 7). (K) Proportions of TCRγδ^+ IELs expressing the CD8αα receptor in young *Ripk1*^ΔCD4 mice (*n* = 4) and *Ripk1*^FL/FL littermates (*n* = 4), measured by flow cytometry. (L) Proportions of TCRγδ^+ IELs expressing the T22 TCR of young *Ripk1*^ΔCD4 mice (*n* = 5) and *Ripk1*^FL/FL littermates (*n* = 8), measured by flow cytometry. (G, L) Data are representative of at least two independent repeats. Data are shown as mean ± SEM, with means being represented by bars or horizontal lines and each dot representing an individual mouse. Statistical significance was calculated in Graphpad Prism by Fisher's LSD two-way ANOVA on Log₂-transformed data (G) CD4^+ and CD8^+ T cells: $p < 0.0001$, Mann-Whitney test (A) $p = 0.6286$, (B) $p = 0.200$, and (E) $p = 0.0317$, or two-sided unpaired T test with Welch's correction (H) $p = 0.039$, (I) $p = 0.0141$, (J) $p = 0.0018$, (K) $p = 0.1345$, (L) $p = 0.7231$. ns = non-significant, *$p < 0.05$, **$p < 0.01$, ****$p < 0.0001$. Young mice: 8–12 weeks old. Aged mice: >6 months old. Source data are available online for this figure.

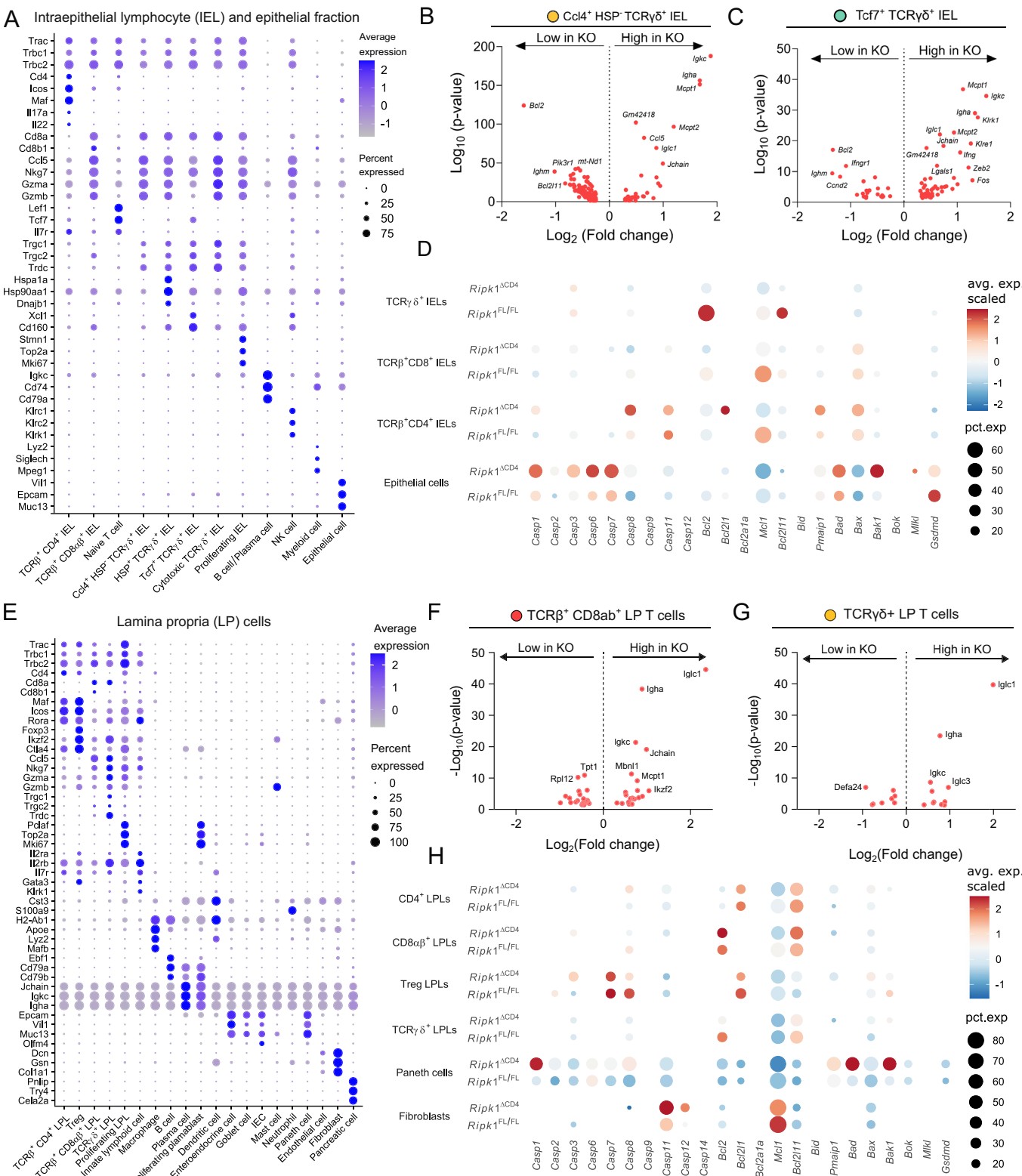

◀ **Figure EV3.   Ripk1^ΔCD4 mice display altered expression of cell death genes in intraepithelial and lamina propria cells.**

(A) Cells were isolated from the epithelial layer (IEL) of young *Ripk1^ΔCD4 and Ripk1^FL/FL* littermates and subjected to single-cell RNA sequencing (scRNA-seq). (B, C) Volcano plots representing significant differentially expressed (DE) genes in the clusters of (B) *Ccl4*+ HSP- TCRγδ+ IELs, and (C) *Tcf7*+ TCRγδ+ IELs of *Ripk1^ΔCD4* mice compared to *Ripk1^FL/FL* littermates. Significance (−Log$_{10}$ of the *p*-value) is indicated on the y-axis, and Log$_2$ of the fold change in gene expression is indicated on the x-axis. (D) Dot plot displaying gene expression of Caspases, intrinsic apoptosis genes, and pore-forming proteins in T cell populations and epithelial cells in the IEL fraction. Size of dots represents the fraction of cells expressing a particular marker, and color intensity indicates mean-normalized scaled expression levels. Cells were isolated from the lamina propria (LP) of young *Ripk1^ΔCD4 and Ripk1^FL/FL* littermates and subjected to single-cell RNA sequencing (scRNA-seq). (E) Dot plot displaying expression of marker genes per cluster used for cluster annotation. The size of dots represents the fraction of cells expressing a particular marker and color intensity indicates mean-normalized scaled expression levels. (F, G) Volcano plots representing significant differentially expressed (DE) genes in the clusters of TCRβ+CD8+ LP T cells (B) and TCRγδ+ LP T cells (C) of *Ripk1^ΔCD4* mice compared to *Ripk1^FL/FL* littermates. Significance (−Log$_{10}$ of the *p*-value) is indicated on the y-axis, and Log$_2$ of the fold change in gene expression is indicated on the x-axis. (H) Dot plot displaying gene expression of Caspases, intrinsic apoptosis genes, and pore-forming proteins in T cell populations, Paneth cells, and Fibroblasts in the LP fraction. Size of dots represents the fraction of cells expressing a particular marker, and color intensity indicates mean-normalized scaled expression levels. Data were obtained from $n = 3$ mice per group (A–H). (B, C, F, G) Wilcox test was used to determine DE genes and *p*-values were adjusted using Bonferroni correction. Young mice: 8–12 weeks old. IEL intraepithelial lymphocytes, LP lamina propria. Source data are available online for this figure.

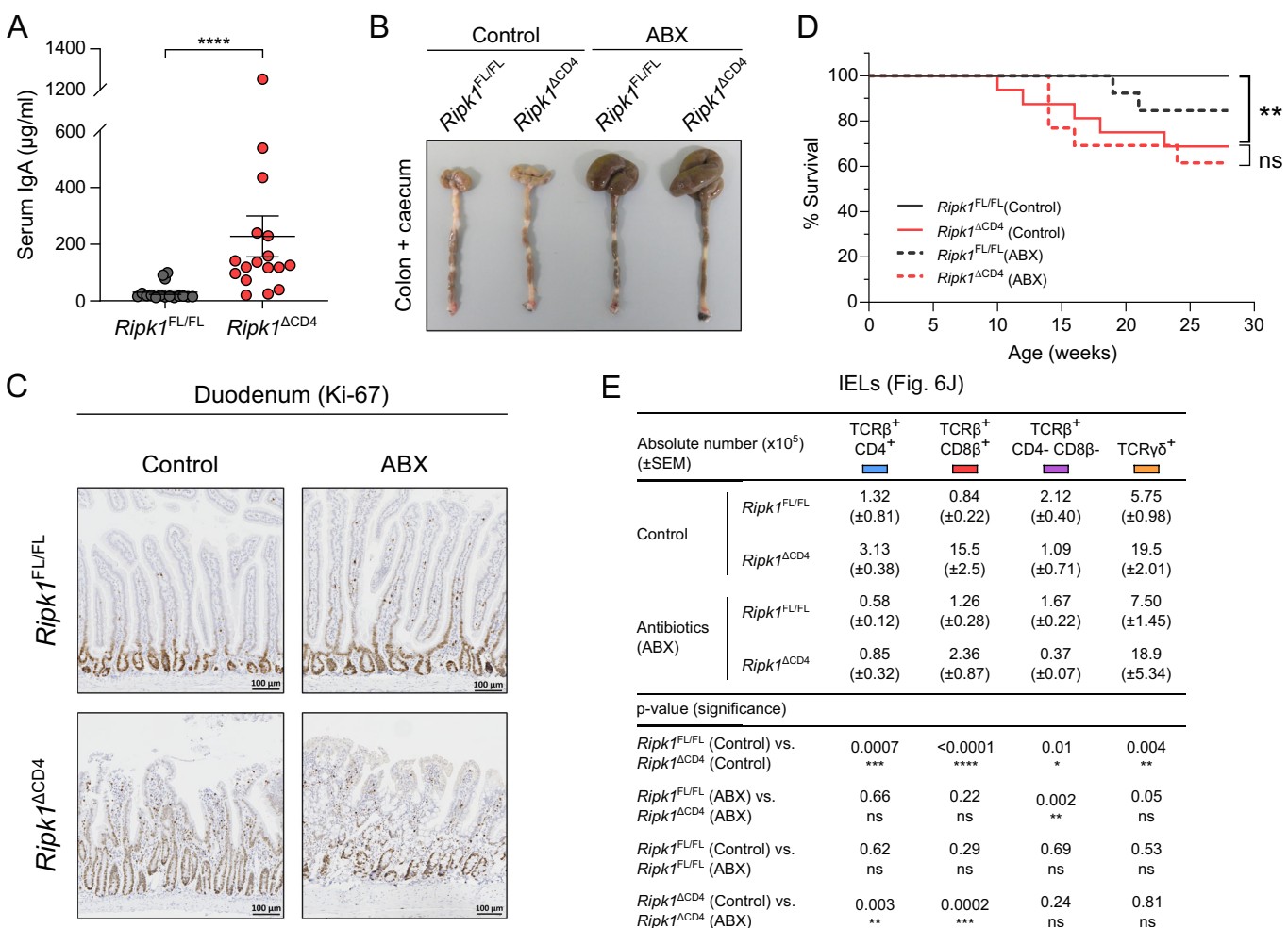

## IELs (Fig. 6J)

| Absolute number (x10⁵) (±SEM) | | TCRβ⁺ CD4⁺ | TCRβ⁺ CD8β⁺ | TCRβ⁺ CD4- CD8β- | TCRγδ⁺ |
|---|---|---|---|---|---|
| Control | Ripk1FL/FL | 1.32 (±0.81) | 0.84 (±0.22) | 2.12 (±0.40) | 5.75 (±0.98) |
| | Ripk1ΔCD4 | 3.13 (±0.38) | 15.5 (±2.5) | 1.09 (±0.71) | 19.5 (±2.01) |
| Antibiotics (ABX) | Ripk1FL/FL | 0.58 (±0.12) | 1.26 (±0.28) | 1.67 (±0.22) | 7.50 (±1.45) |
| | Ripk1ΔCD4 | 0.85 (±0.32) | 2.36 (±0.87) | 0.37 (±0.07) | 18.9 (±5.34) |

| p-value (significance) | TCRβ⁺ CD4⁺ | TCRβ⁺ CD8β⁺ | TCRβ⁺ CD4- CD8β- | TCRγδ⁺ |
|---|---|---|---|---|
| Ripk1FL/FL (Control) vs. Ripk1ΔCD4 (Control) | 0.0007 *** | <0.0001 **** | 0.01 * | 0.004 ** |
| Ripk1FL/FL (ABX) vs. Ripk1ΔCD4 (ABX) | 0.66 ns | 0.22 ns | 0.002 ** | 0.05 ns |
| Ripk1FL/FL (Control) vs. Ripk1FL/FL (ABX) | 0.62 ns | 0.29 ns | 0.69 ns | 0.53 ns |
| Ripk1ΔCD4 (Control) vs. Ripk1ΔCD4 (ABX) | 0.003 ** | 0.0002 *** | 0.24 ns | 0.81 ns |

**Figure EV4.  SI elongation in *Ripk1*ᐞCD4 mice is a response to the intestinal microbiome, while villus atrophy and TCRγδ⁺ IEL expansion are not.**

(A) Serum IgA concentrations in aged *Ripk1*ΔCD4 mice (*n* = 17) and *Ripk1*FL/FL littermates (*n* = 16), measured by multiplex analysis (Meso Scale Discovery). Data are shown as mean ± SEM, with means represented by bars and each dot representing an individual mouse. (B) Representative images of the colon with caecum isolated from aged *Ripk1*ΔCD4 mice and *Ripk1*FL/FL littermates, with and without ABX treatment. (C) Ki-67 staining on the duodenum of aged *Ripk1*ΔCD4 mice and *Ripk1*FL/FL littermates, with and without ABX treatment. Images are representative of two independent repeats with (*n* = 3) *Ripk1*ΔCD4 mice (both ABX and control groups) or (*n* = 4) *Ripk1*FL/FL littermates (both ABX and control groups). (D) Kaplan–Meier survival analysis of *Ripk1*ΔCD4 mice (*n* = 16) and *Ripk1*FL/FL littermates (*n* = 21) without ABX control, and *Ripk1*ΔCD4 mice (*n* = 13) and *Ripk1*FL/FL mice (*n* = 13) on ABX. (E) Table corresponding to Fig. 6J, indicating the absolute numbers of the different IEL populations and resulting *p*-values of the statistical analysis (Fisher's LSD two-way ANOVA), with (*n* = 4) *Ripk1*ΔCD4 mice (control) (*n* = 3) *Ripk1*ΔCD4 mice (ABX) or (*n* = 5) *Ripk1*FL/FL littermates (both ABX and control groups). (A, D) Data are combined from two independent repeats, or (B, C, E) representative of two independent repeats. Statistical significance was calculated in Graphpad Prism by (A) Mann-Whitney test *p* < 0.0001, (D) Gehan-Wilcoxon test for survival, *Ripk1*FL/FL vs *Ripk1*ΔCD4 (Control): *p* = 0.0067, Control vs ABX (*Ripk1*ΔCD4): *p* = 0.8301, or (E) two-way ANOVA on Log₂-transformed data with *p*-values indicated in table. ns = non-significant, *\*p* < 0.05, *\*\*p* < 0.01, *\*\*\*p* < 0.001, *\*\*\*\*p* < 0.0001. Aged mice: >6 months old. IEL intraepithelial lymphocytes. Source data are available online for this figure.

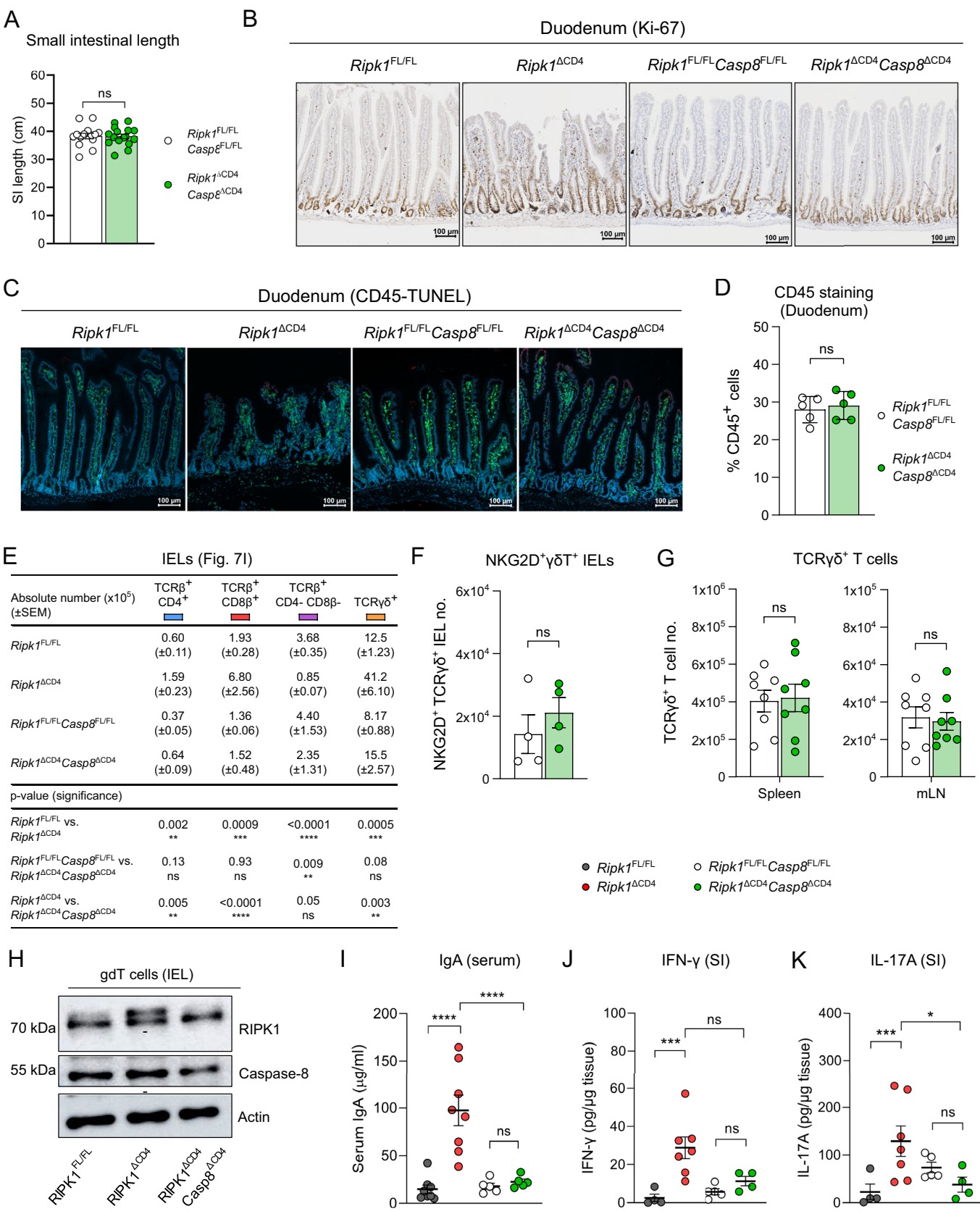

◀  **Figure EV5.  Villus atrophy is TNFR1-dependent, and T cell ablation of caspase-8 prevents the development of intestinal pathology.**

(A) Quantification of the absolute small intestine (SI) length of $Ripk1^{\Delta CD4}Casp8^{\Delta CD4}$ mice ($n = 16$) and $Ripk1^{FL/FL}Casp8^{FL/FL}$ littermates ($n = 14$). (B) Ki-67 staining on duodenum sections of aged $Ripk1^{\Delta CD4}Casp8^{\Delta CD4}$ mice and $Ripk1^{FL/FL}Casp8^{FL/FL}$ littermates. (C) CD45-TUNEL staining on duodenum sections of aged $Ripk1^{\Delta CD4}$ and $Ripk1^{\Delta CD4}Casp8^{\Delta CD4}$ mice and their respective $Ripk1^{FL/FL}$ and $Ripk1^{FL/FL}Casp8^{FL/FL}$ littermates (CD45 = Green; TUNEL = Red). (D) Quantification of CD45 staining in the duodenum of aged $Ripk1^{\Delta CD4}Casp8^{\Delta CD4}$ mice and $Ripk1^{FL/FL}Casp8^{FL/FL}$ littermates ($n = 5$), represented as the percentage of CD45-positive cells within the total number of cells (determined by DAPI staining). (E) Table corresponding to Fig. 7I, indicating the absolute numbers of the different IEL populations and their resulting $p$-values of the statistical analysis (Fisher's LSD two-way ANOVA). (F) Absolute numbers of NKG2D$^+$ TCRγδ$^+$ IELs in the SI epithelial layer of young $Ripk1^{\Delta CD4}Casp8^{\Delta CD4}$ mice and $Ripk1^{FL/FL}Casp8^{FL/FL}$ littermates ($n = 4$), measured by flow cytometry. (G) Absolute numbers of TCRγδ$^+$ T cells in the spleen (left) and mLN (right) of young $Ripk1^{\Delta CD4}Casp8^{\Delta CD4}$ mice and $Ripk1^{FL/FL}Casp8^{FL/FL}$ littermates ($n = 8$), measured by flow cytometry. (H) Western blot on FACS-sorted TCRγδ$^+$ IELs, showing RIPK1, Caspase-8, and Actin in $Ripk1^{FL/FL}$, $Ripk1^{\Delta CD4}$ and $Ripk1^{\Delta CD4}Casp8^{\Delta CD4}$ mice. (I) Serum IgA concentrations in young $Ripk1^{\Delta CD4}$ mice ($n = 8$) and $Ripk1^{FL/FL}$ littermates ($n = 8$), and aged $Ripk1^{\Delta CD4}Casp8^{\Delta CD4}$ mice ($n = 5$) and $Ripk1^{FL/FL}Casp8^{FL/FL}$ littermates ($n = 5$), measured by multiplex analysis (Meso Scale Discovery). (J, K) Tissue concentrations of (J) IFN-γ, and (K) IL-17A in the SI of young $Ripk1^{\Delta CD4}$ ($n = 7$) and $Ripk1^{\Delta CD4}Casp8^{\Delta CD4}$ ($n = 5$) mice and their respective $Ripk1^{FL/FL}$ ($n = 4$) and $Ripk1^{FL/FL}Casp8^{FL/FL}$ ($n = 5$) littermates, measured by multiplex analysis (Meso Scale Discovery). Data are representative of two (E) or three (A) independent repeats, or (G) are combined from two independent repeats. Data are shown as mean ± SEM, with means being represented by bars or horizontal lines and each dot representing an individual mouse. Statistical significance was calculated in Graphpad Prism by unpaired T-test with Welch's correction (A, D, F, G), or Fisher's LSD two-way ANOVA on Log$_2$-transformed data (E) with p-values indicated in table (I) $Ripk1^{FL/FL}$ vs $Ripk1^{\Delta CD4}$: $p < 0.0001$, $Ripk1^{\Delta CD4}$ vs $Ripk1^{\Delta CD4}Casp8^{\Delta CD4}$: $p < 0.0001$, $Ripk1^{FL/FL}Casp8^{FL/FL}$ vs $Ripk1^{\Delta CD4}Casp8^{\Delta CD4}$: $p = 0.7831$; (J) $Ripk1^{FL/FL}$ vs $Ripk1^{\Delta CD4}$: $p = 0.0002$, $Ripk1^{\Delta CD4}$ vs $Ripk1^{\Delta CD4}Casp8^{\Delta CD4}$: $p = 0.1071$, $Ripk1^{FL/FL}Casp8^{FL/FL}$ vs $Ripk1^{\Delta CD4}Casp8^{\Delta CD4}$: $p = 0.1931$; (K) $Ripk1^{FL/FL}$ vs $Ripk1^{\Delta CD4}$: $p = 0.0009$, $Ripk1^{\Delta CD4}$ vs $Ripk1^{\Delta CD4}Casp8^{\Delta CD4}$: $p = 0.0401$, $Ripk1^{FL/FL}Casp8^{FL/FL}$ vs $Ripk1^{\Delta CD4}Casp8^{\Delta CD4}$: $p = 0.1621$. ns = non-significant, *$p < 0.05$, **$p < 0.01$, ***$p < 0.001$, ****$p < 0.0001$. Young mice: 8–12 weeks old, aged mice: >6 months old. IEL intraepithelial lymphocytes. Source data are available online for this figure.

