## [Peer Review File · EMBO Reports]

RIPK1 ablation in T cells results in spontaneous enteropathy and TNF-driven villus atrophy

Jelle Huysentruyt, Wolf Steels, Mario Ruiz Pérez, Bruno Verstraeten, Tatyana Divert, Kayleigh Flies, Kelly Lemeire, Nozomi Takahashi, Elke De Bruyn, Marie Joossens, Andrew S. Brown, Bart N. Lambrecht, Wim Declercq, Tom Vanden Berghe, Jonathan Maelfait, Peter Vandenabeele, and Peter Tougaard

Corresponding author(s): Peter Vandenabeele (PeterJV.Vandenabeele@UGent.be)

Review Timeline:

Submission Date:	9th Jun 23
Editorial Decision:	31st Jul 23
Revision Received:	18th Dec 24
Editorial Decision:	14th Feb 25
Revision Received:	11th Mar 25
Accepted:	18th Mar 25

Editor: Achim Breiling

Transaction Report:

Dear Prof. Vandenabeele,

Thank you for the submission of your manuscript to EMBO reports. I have now received the reports from the three referees that were asked to evaluate your study, which can be found at the end of this message.

As you will see, the referees think that these findings are of high interest. However, they have several comments, concerns, and suggestions to improve the manuscript, indicating that a major revision is necessary to allow publication of the study in EMBO reports. As the reports are below, and all the referee concerns need to be addressed, I will not detail them here.

Given the constructive referee comments, I would like to invite you to revise your manuscript with the understanding that all referee concerns must be addressed in the revised manuscript and in a detailed point-by-point response. Acceptance of your manuscript will depend on a positive outcome of a second round of review. It is EMBO reports policy to allow a single round of revision only and acceptance of the manuscript will therefore depend on the completeness of your responses included in the next, final version of the manuscript.

- 1) a .docx formatted version of the final manuscript text (including legends for main figures, EV figures and tables), but without the figures included. Figure legends should be compiled at the end of the manuscript text.
- 2) individual production quality figure files as .eps, .tif, .jpg (one file per figure), of main figures (up to 8) and EV figures (up to 5). Please upload these as separate, individual files upon re-submission.

- 4) a complete author checklist, which you can download from our author guidelines (<https://www.embopress.org/page/journal/14693178/authorguide>). Please insert page numbers in the checklist to indicate where the requested information can be found in the manuscript. The completed author checklist will also be part of the RPF.

- 5) that primary datasets produced in this study (e.g. RNA-seq, ChIP-seq, structural and array data) are deposited in an

appropriate public database. If no primary datasets have been deposited, please also state this in a dedicated section (e.g. 'No primary datasets have been generated and deposited'), see below.

The accession numbers and database should be listed in a formal "Data Availability" section (placed after Materials & Methods) that follows the model below. This is now mandatory (like the COI statement). Please note that the Data Availability Section is restricted to new primary data that are part of this study. This section is mandatory. As indicated above, if no primary datasets have been deposited, please state this in this section

Data availability

8) Regarding data quantification and statistics, please make sure that the number "n" for how many independent experiments were performed, their nature (biological versus technical replicates), the bars and error bars (e.g. SEM, SD) and the test used to calculate p-values is indicated in the respective figure legends (also for potential EV figures and all those in the final Appendix). Please also check that all the p-values are explained in the legend, and that these fit to those shown in the figure. Please provide statistical testing where applicable. Please avoid the phrase 'independent experiment', but clearly state if these were biological or technical replicates. Please also indicate (e.g. with n.s.) if testing was performed, but the differences are not significant. In case n=2, please show the data as separate datapoints without error bars and statistics. See also: <http://www.embopress.org/page/journal/14693178/authorguide#statisticalanalysis>

9) Please add scale bars of similar style and thickness to microscopic images, using clearly visible black or white bars (depending on the background). Please place these in the lower right corner of the images themselves. Please do not write on or near the bars in the image but define the size in the respective figure legend.

10) Please also add size markers (rulers) to mouse, organ or tumor images.

11) Please also note our reference format:

12) We now use CRediT to specify the contributions of each author in the journal submission system. CRediT replaces the author contribution section. Please use the free text box to provide more detailed descriptions and do not provide your final manuscript text file with an author contributions section. See also our guide to authors: <https://www.embopress.org/page/journal/14693178/authorguide#authorshipguidelines>

13) We would encourage you to use 'Structured Methods', our new Materials and Methods format. According to this format, the Materials and Methods section should include a Reagents and Tools Table (listing key reagents, experimental models, software and relevant equipment and including their sources and relevant identifiers) followed by a Methods and Protocols section in which we encourage the authors to describe their methods using a step-by-step protocol format with bullet points, to facilitate the adoption of the methodologies across labs. More information on how to adhere to this format as well as downloadable templates (.doc or .xls) for the Reagents and Tools Table can be found in our author guidelines (section 'Structured Methods'):

14) Please add up to 5 keywords to the manuscript text and order the manuscript sections like this, using these names: Title page - Abstract - Keywords - Introduction - Results - Discussion - Materials and Methods - Data availability section - Acknowledgements - Disclosure and Competing Interests Statement - References - Figure legends - Expanded View Figure legends

I look forward to seeing a revised version of your manuscript when it is ready. Please let me know if you have questions or comments regarding the revision.

Yours sincerely,

Referee #1:

In this study Huysentruyt and colleagues investigated the impact of RIPK1 deletion in T cells using CD4 promoter-driven Cre-mediated deletion on the frequency of peripheral and intestinal T cells, and the development of intestinal pathologies. The finding that RIPK1 deletion results in reduced peripheral T cells, in line with a protective role of RIPK1 in T cell death, but unexpectedly an accumulation of unconventional intraepithelial lymphocytes (IEL) in the intestinal epithelium, especially in the duodenum, which was associated with increased IFN γ production. They further observe elongation of the small intestine and crypt atrophy, which was corrected upon treatment of mice with antibiotics, suggesting a role of an anti-bacterial immune response. In contrast, the increase in TCR γ d IEL was not affected by antibiotics treatment, but only if caspase 8 was co-deleted with RIPK1 in T cells.

This is an interesting and mostly well performed study clearly supporting an important role for RIPK1 in T cell development, homeostasis and T cell regulated intestinal inflammation. However, in the view of this reviewer some of data may be wrongly interpreted. Various analyses and approaches are suggested to confirm/disprove these alternative interpretations.

Specific comments:

1) It has been described in the past that autoreactive TCR γ d T cells escape negative selection in the thymus at early ages and find sanctuary in the intestinal mucosa (e.g. PMID: 10545528). There, they do normally not exhibit an autoreactive phenotype, because they are probably controlled by regulatory T cells (Tregs) or may have a regulatory function themselves. In the likely absence of thymus-derived CD4⁺ FoxP3⁺ Tregs in RIPK1-deficient mice these autoreactive T cells could massively expand and result in the pathology noticed. The authors should analyze whether these TCR γ d IEL have such autoreactive TCRs. The authors should also check whether these are CD8 α ⁺ and not CD8 α ⁺. Furthermore, it would be interesting to see whether these IEL show a certain clonality of the TCR.

2) The authors mention that they observe CD4⁺ FoxP3⁺ T cells in the gut. They should isolate them or maybe also other unconventional regulatory T cells from the gut of wt and RPK1 KO mice and analyze their suppressive potential. It may well be that the problem is more the lack of T cell suppression than the selective induction of γ d T cells. They authors should further see whether they can induce Tregs in CD4⁺ T cells by TGF β in both mouse lines and whether they are suppressive. Furthermore it would be interesting to see whether adoptive transfer of CD4⁺ CD45Rb^{lo} T cells from wild type but not KO mice are capable to rescue the pathology in CD4-RIPK1 mice. This would indicate that the defect is mostly due to the Tregs and not the effector t cells.

3) It is strange that the pathology is mostly concentrated to the duodenum, given that in most other inflammatory pathologies of the gut, the colon is mostly affected due to the bacterial burden. What could be the reason? Again, this could point towards an autoimmune disease rather than anti-bacterial response. Along these lines ABX did not affect TCR γ d T cells numbers. The anti-bacterial response could be a consequence of the tissue destruction in the duodenum rather than a cause. Along these lines

increased TUNEL pos cells are seen at the lower part of the villus.

4) There are multiple labels missing in various figures, e.g. Fig 2 A, C, G, Fig 3 G,

5) Fig 7: Can the author explain why there are differences in the total T cell number and quantification of SI length between the two control groups (Ripk1^{FL/FL} and Ripk1^{FL/FL} Casp8^{FL/FL})?

minor:

- the authors should use the term IEL not IET

- 46: results

- 65: T cells

- 67: the lung

- 153f: references to Fig. 3 B also in the next sentence:

It might be helpful to put arrows behind the statistic and legend of Fig 3 B and C to indicate which cell population is increasing or decreasing or not changing

- 199ff, 226: Consistency: write cluster or C

- 426: T cell-specific

- 228: (Fig. 5 B)

- 698, 752, 844: epithelial layer (IET): It is not clear why IET is in brackets

- 684: IET (top) or LP T cell (bottom) population

- 710, 735: Data were obtained from three mice per group but only one column displayed in the heatmap which should represent individual samples (Figure 4 and 5)

- 767-769: Figure 7 legend C and D does not match with Figure 7. C = Quantification of SI and D = absolute CD4⁺ and CD8⁺ T cell numbers

- 821, 885: no data of young mice shown, can be omitted

- 854, 866: no data of aged mice shown, can be omitted

Referee #2:

Huysentruyt et al. present a manuscript that describes a striking intestinal phenotype in conditional knockout mice lacking Ripk1 specifically in T cells. Ripk1 is a central regulator of cell death and inflammation and linked to the activation of the pro-inflammatory transcription factor NF- κ B as well as to the induction of caspase-8-dependent apoptosis or Ripk3/MLKL-mediated necroptosis. Although T cell-specific knockout mice of Ripk1 have been described by some of the current authors and others before, this study goes a significant step further and analyzes not only young but also aged mice. Surprisingly, these mice have an elongated small intestine and exhibit gut inflammation - including villus atrophy - which appears to be driven by the expansion of $\gamma\delta$ T cells. Of note, the elongated intestine depends on the microbiota since it can be reverted by antibiotics treatment. In contrast, villus atrophy and $\gamma\delta$ T cell expansion is not affected by antibiotics. Yet, all phenotypes are rescued by crossing in a floxed caspase-8 allele suggesting that Ripk1 protects T cells from caspase8-mediated apoptosis ensuring gut homeostasis. These findings are novel and important. I just have a few comments.

The caspase-8 co-deletion suggests a role for Ripk1 in T cell apoptosis. However, caspase-8 also has nonapoptotic functions. Therefore, the genetic approach is not a final proof that apoptosis plays a role here. Do the authors detect enhanced cell death in the intestines of their mice? For instance, is TUNEL staining increased in sections of the gut? Although the stainings have been performed, I am missing a description of these results.

On page 13/14 the authors discuss the role of TNF in gut inflammation. Could TNF play a role here in their system?

According to the scRNAseq data, Bcl2 expression is reduced in the conditional knockout mice. However, this applies to both, TCR β ⁺ CD8⁺ and TCR $\gamma\delta$ ⁺ populations, which are reduced and expanded, respectively. Is Bcl2 really the important molecule here? What about other cell death-related genes?

The expansion of $\gamma\delta$ T cells in the conditional knockout mice is impressive. However, the question arises what could be the driver of this $\gamma\delta$ T cell expansion? After all, Ripk1 deletion happens in conventional T cells due to CD4-Cre driver. What is the authors idea on this phenotype?

The antibiotics treatment strongly indicates a crucial role of the microbiota in the observed phenotype. However, there is a hen-and-egg problem: is the microbiota changed due to the ongoing inflammation or vice versa? Although addressing this question is clearly beyond the scope of the current study, the question remains as to why the intestinal microbiome is altered upon Ripk1 deletion in T cells? Can the authors at least speculate how Ripk1 deficiency in T cells impacts on the microbiome?

Type 3 immune response not necessarily anti-bacterial. Its primary target are extracellular pathogens such as certain bacteria and fungi. This should be corrected.

Minor points:

Lane 380: A space is missing at "...T cells.While..."

Lane 505: A space is missing at "...2mM EDTA..."

Lane 580: A space is missing at "...3nM..."

Figure 1H: the labeling K45A is missing.

Figures 2, S2, S6 and S7: Please indicate in which color CD45 and TUNEL are labeled. This is not indicated in the figure, the legends or the materials and methods section.

Referee #3:

- You should be trying to help the work get published not necessarily in this journal but ultimately.
- Don't criticize an experiment unless you can tell the authors how they could do it better. "If you just want to throw darts," he would say, "go to the pub."
- Keep in mind that no one ever built a statue to a critic.
- Try to act as a peer in the process of peer review.

Science Signaling 2009 Michael Yaffe

Title: RIPK1 ablation results in T cell apoptosis driving small intestinal inflammation in mice

Manuscript # EMBOR-2023-57617V1

General Remarks:

This study examines the phenotype of mice with a RIPK1 deficiency in conventional T-cells. The authors show that the mice develop a chronic wasting syndrome and intestinal pathology that is primarily confined to the duodenum. The phenotype is not observed in RIPK1 kinase dead mice, and concomitant ablation of caspase-8 prevented the pathology. This clearly shows that the phenotype depends on the scaffold function of RIPK1 and that it works by preventing apoptosis. This mirrors the phenotype of human patients and a question that could be addressed in the discussion is why whole body dysfunction in humans seems to recapitulate this cell type specific deletion in mice?

Since there is a clear increase in the TCR $\gamma\delta$ T population in the Δ CD4 RIPK1 knock out mice, it would add to the insights from the manuscript if the authors could show that these cells either don't have RIPK1 expressed or are insensitive to TNF induced cell death, unlike their other T-cell counterparts.

The manuscript is well written and presented, the statistics properly described, good numbers of mice are analysed.

Specific Remarks

A quibble, but line 52, can a D325 mutation in RIPK1 properly be described as a gain of function? It loses the ability to be cleaved by caspase-8 and the mutant cells are now more sensitive to cell death, but one could therefore to my mind equally claim that it is a loss of function.

Rebuttal EMBOR-2023-57617V1.

RIPK1 ablation in T cells results in spontaneous enteropathy and TNF-driven villus atrophy

Dear Editor,
Dear Dr. Achim Breiling,

Thank you very much for your editorial work.

We have now responded to the different items raised by the referees. Although we realize that we have gone far beyond the deadline, we would still ask you to reconsider our manuscript and process with the rebuttal process. This delay has been due to several reasons (additional experiments and animal breeding, people have left the unit).

I am convinced that the rebuttal process has resulted in a stronger manuscript. We have generated new Figures and figure panels. Changes in the manuscript are highlighted in yellow.

Please find our point-by-point reply to the reviewers below, marked by the letters a-x. Explanatory figures can be found at the end of the text. We have highlighted the new figure panels.

Kind regards,

Peter Vandenabeele

Referee #1:

In this study Huysentruyt and colleagues investigated the impact of RIPK1 deletion in T cells using CD4 promoter-driven Cre-mediated deletion on the frequency of peripheral and intestinal T cells, and the development of intestinal pathologies. The find that RIPK1 deletion results in reduced peripheral T cells, in line with a protective role of RIPK1 in T cell death, but unexpectedly an accumulation of unconventional intraepithelial lymphocytes (IEL) in the intestinal epithelium, especially in the duodenum, which was associated with increased IFN γ production. They further observe elongation of the small intestine and crypt atrophy, which was corrected upon treatment of mice with antibiotics, suggesting a role of an anti-bacterial immune response. In contrast, the increase in TCR gd IEL was not affected by antibiotics treatment, but only if caspase 8 was co-deleted with RIPK1 in T cells.

This is an interesting and mostly well performed study clearly supporting an important role for RIPK1 in T cell development, homeostasis and T cell regulated intestinal inflammation. However, in the view of this reviewer some of data may be wrongly interpreted. Various analyses and approaches are suggested to confirm/disprove these alternative interpretations.

Specific comments:

1) It has been described in the past that autoreactive TCR gd T cells escape negative selection in the thymus at early ages and find sanctuary in the intestinal mucosa (e.g. PMID: 10545528). There, they do normally not exhibit an autoreactive phenotype, because they are probably controlled by regulatory T cells (Tregs) or may have a regulatory function themselves. In the likely absence of thymus-derived CD4⁺ FoxP3⁺ Tregs in RIPK1-deficient mice these autoreactive T cells could massively expand and result in the pathology noticed. the authors should analyze whether these TCRgd IEL have such autoreactive TCRs. The authors should also check whether these are CD8 $\alpha\alpha$ ⁺ and not CD8 $\alpha\beta$ ⁺. Furthermore, it would be interesting to see whether these IEL show a certain clonality of the TCR.

a) There are indeed several options for determining the TCR $\alpha\beta$ clonality in mice, we were not able to find any commercially available way of determining the TCR $\gamma\delta$ clonality in mice. Furthermore, since the scRNAseq data were performed with 3' rather than 5' sequencing, we were not able to investigate the TCR clonality in the single cell data. However, TCR tetramer stainings were performed to determine whether the $\gamma\delta$ T cells in the intestinal mucosa were more enriched for the autoreactive T22-TCRs as mentioned by the reviewer in (PMID: 10545528). We did not find any enrichment of these autoreactive $\gamma\delta$ TCRs between the genotypes and included these data as an extra panel in the Extended view (EV) Figure (Fig. EV2L**).**

In Ripk1^{ACD4} mice, we observed increased numbers of $\gamma\delta$ T cells in all compartments investigated compared to littermate controls (Figs. 3 B-D**). We have included a graph showing the abundance $\gamma\delta$ T IELs expressing the CD8 $\alpha\alpha$ homodimer which was unchanged between genotypes (**Fig. EV2K**). Since the total number of $\gamma\delta$ T cells were increased the total number of CD8 $\alpha\alpha$ ⁺ $\gamma\delta$ T IELs were also increased, while not expressing notable levels of the CD8 $\alpha\beta$ co-receptor (**Rebuttal Figure 1**).**

2) The authors mention that they observe CD4⁺ FoxP3⁺ T cells in the gut. They should isolate them or maybe also other unconventional regulatory T cells from the gut of wt and RPK1 KO mice and analyze their suppressive potential. It may well be that the problem is more the lack of T cell suppression that the selective induction of gd T cells. They authors

should further see whether they can induce Tregs in CD4⁺ T cells by TGF beta in both mouse lines and whether they are suppressive. Furthermore it would be interesting to see whether adoptive transfer of CD4⁺ CD45Rblo T cells from wild type but not KO mice are capable to rescue the pathology in CD4-RIPK1 mice. This would indicate that the defect is mostly due to the Tregs and not the effector t cells.

b) Thanks for this remark, which we have addressed experimentally. We observed increased numbers of FoxP3⁺ Treg cells in the intestinal mucosa of *Ripk1*^{ΔCD4} mice between the genotypes, the number of FoxP3⁺ Treg cells remained unchanged in the lamina propria (**Fig. 3H**). The enhanced or unchanged numbers of SI FoxP3⁺ Tregs in the *Ripk1*^{ΔCD4} mice either means that FoxP3⁺ Tregs are not involved in the specific pathology or their functionality is impaired. We show that the microbiota induced the enhanced Foxp3⁺ Tregs in the intestinal mucosa of *Ripk1*^{ΔCD4} mice (**Fig. 6I**), but despite this microbiota-driven Foxp3⁺ Tregs infiltration, mice treated with broad-spectrum antibiotics still displayed severe intestinal pathology (**Fig. 6**). These data indicate the increased numbers Foxp3⁺ Tregs are independent of the underlying drivers of the intestinal pathology.

Additionally, we investigated the production of anti-inflammatory cytokines in the CD4 T cell population to investigate their regulatory functions and found that while the total TGFβ levels and TGFβ-producing lamina propria CD4 T cells remained unchanged, IL-10-producing CD4 T cells were reduced in the *Ripk1*^{ΔCD4} mice (**Figs. 3 I-J, EV1H**). The reduction of IL-10-producing CD4 T cells in *Ripk1*^{ΔCD4} mice might explain the increased tissue levels of TNF and IFNγ observed in *Ripk1*^{ΔCD4} mice (**Fig. 1G**).

3) It is strange that the pathology is mostly concentrated to the duodenum, given that in most other inflammatory pathologies of the gut, the colon is mostly affected due to the bacterial burden. What could be the reason? Again, this could point towards an autoimmune disease rather than anti-bacterial response. Along these lines ABX did not affect TCR gd T cells numbers. The anti-bacterial response could be a consequence of the tissue destruction in the duodenum rather than a cause. Along these lines increased TUNEL pos cells are seen at the lower part of the villus.

c) This is a great question and as the reviewer mentions the villus atrophy and γδT cell infiltration was not rescued by the use of life-long antibiotics although the intestinal elongation was (**Figs. 6 B, H and J**), revealing that the bacterial burden only is responsible for part of the observed pathology. γδT cells have the highest abundance in the duodenum compared to jejunum and ileum and this can be observed in both SPF and germ free mice (PMID: 28942917). Additionally, we found that the intestinal γδT cells displayed a more inflammatory phenotype with enhanced TNF production. Furthermore, we determined that the villus atrophy could be blocked by crossing the *Ripk1*^{ΔCD4} mice with TNFR1^{-/-} knockout mice (**Figs. 7 K-M**). This has been elaborated in the discussion.

Lines 380-393: “In Ripk1^{ΔCD4} mice, certain features, such as crypt hyperplasia and intestinal elongation, were microbiota-dependent, whereas villus atrophy persisted despite broad-spectrum antibiotic treatment. The duodenal crypt hyperplasia, the intestinal elongation, and the expansion of Tregs, CD4⁺ and CD8β⁺ TCRβ⁺ IELs were all rescued following antibiotics, suggesting that these phenotypes represent a response towards the intestinal microbiome. The expansion of TCRγδ⁺ IELs and the villus atrophy remained in antibiotics-treated Ripk1^{ΔCD4} mice, indicating that these symptoms were independent of the microbiome alterations. Comparing germ-free and specific pathogen-free (SPF) mice has also previously shown that

the total TCR $\gamma\delta^+$ IEL numbers are relatively unaffected by the presence of the microbiota (Bandeira et al., 1990; Hoytema van Konijnenburg et al., 2017). TCR $\gamma\delta^+$ IELs may affect the microbial composition through the expression of antimicrobial peptides (Rezende et al., 2023). However, since the antibiotic treatment did not affect the villus atrophy or increased mortality in the Ripk1^{ACD4} mice, the microbiota is unlikely to be the underlying cause of these pathologies.

Lines 428-435: *“Furthermore, in Crohn’s disease, TCR $\gamma\delta^+$ IELs promote pathology by inducing TNF-mediated shedding of IECs, leading to villus atrophy in areas with active disease (Hu et al., 2022). Given the increased number of TNF-producing TCR $\gamma\delta^+$ IELs and enhanced tissue concentrations of TNF, we crossed the Ripk1^{ACD4} mice onto a Tnfr1^{-/-} background. Ripk1^{ACD4}Tnfr1^{-/-} mice were protected from villus atrophy, yet they still exhibited microbiome-mediated intestinal elongation and crypt hyperplasia. This suggests that TNF signaling plays a critical role in mediating villus atrophy, while other pathways remain active in tissue remodeling.*”

We have quantified TUNEL positive cells in the small intestine and found that there is an increased number of TUNEL positive cells in the duodenum of Ripk1^{ACD4} mice compared to wild-type littermates (**Fig. EV2E**). The TUNEL positive cells were mostly confined to the epithelium at the top of the villi (**Fig. 2G**). Along these lines, we observed upregulation of gene expression associated with stress and ROS responses in SI epithelial cells (**Figs. 4F and 4G**), indicating that the intestinal pathology in the Ripk1^{ACD4} mice is associated with the viability of the epithelial cells.

4) There are multiple labels missing in various figures, e.g. Fig 2 A, C, G, Fig 3 G,

d) These have been corrected.

5) Fig 7: Can the author explain why there are differences in the total T cell number and quantification of SI length between the two control groups (Ripk1FL/FL and Ripk1FL/FL Casp8FL/FL)?

e) The difference mentioned are likely due to variation between these two floxed mouse lines. Although the two mouse lines were related, they have been bred separately for a while. To make the right comparisons, we made sure to always use littermate controls. This has been explicitly stated in the Materials and Methods section.

Lines 504-506: *“Experimental mouse lines were cohoused, and littermate controls were used in all experiments to ensure appropriate comparisons.”*

minor:

- the authors should use the term IEL not IET $\uparrow\downarrow$

f) This has been adapted throughout the manuscript.

- 46: results

- 65: T cells
- 67: the lung

g) These points have been corrected.

- 153f: references to Fig. 3 B also in the next sentence:

It might be helpful to put arrows behind the statistic and legend of Fig 3 B and C to indicate which cell population is increasing or decreasing or not changing

h) Arrows have now been added to Figure 3 B and 3 C to indicate the changing populations.

- 199ff, 226: Consistency: write cluster or C

i) The cluster numbers have all been removed and changed to cell type names.

- 426: T cell-specific
- 228: (Fig. 5 B)
- 698, 752, 844: epithelial layer (IET): It is not clear why IET is in brackets
- 684: IET (top) or LP T cell (bottom) population

j) These remarks have been corrected.

- 710, 735: Data were obtained from three mice per group but only one column displayed in the heatmap which should represent individual samples (Figure 4 and 5)

k) We have adapted the analyses of the scRNAseq in Figure 4 and 5 quite substantially and do not show the heatmaps anymore, although the differences are still shown in the volcano plots. Instead we show new pathway analyses and an extensive dot plot analysis with specific focus on different cell death genes as suggested by Reviewer 2 (Figs. EV3D and EV3H). The new analysis is quite extensive and is easier to interpret when the samples are combined. The three individual samples per group in the scRNAseq data are now explicitly illustrated by individual dots in Fig. 4 B.

- 767-769: Figure 7 legend C and D does not match with Figure 7. C = Quantification of SI and D = absolute CD4+ and CD8+ T cell numbers

- 821, 885: no data of young mice shown, can be omitted

- 854, 866: no data of aged mice shown, can be omitted

l) These remarks have been adapted.

Referee #2:

Huysentruyt et al. present a manuscript that describes a striking intestinal phenotype in conditional knockout mice lacking Ripk1 specifically in T cells. Ripk1 is a central regulator of cell death and inflammation and linked to the activation of the pro-inflammatory transcription factor NF- κ B as well as to the induction of caspase-8-dependent apoptosis or Ripk3/MLKL-mediated necroptosis. Although T cell-specific knockout mice of Ripk1 have been described by some of the current authors and others before, this study goes a significant step further and analyzes not only young but also aged mice. Surprisingly, these mice have an elongated small intestine and exhibit gut inflammation - including villus atrophy - which appears to be driven by the expansion of $\gamma\delta$ T cells. Of note, the elongated intestine depends on the microbiota since it can be reverted by antibiotics treatment. In contrast, villus atrophy and $\gamma\delta$ T cell expansion is not affected by antibiotics. Yet, all phenotypes are rescued by crossing in a floxed caspase-8 allele suggesting that Ripk1 protects T cells from caspase-8-mediated apoptosis ensuring gut homeostasis. These findings are novel and important. I just have a few comments.

The caspase-8 co-deletion suggests a role for Ripk1 in T cell apoptosis. However, caspase-8 also has nonapoptotic functions. Therefore, the genetic approach is not a final proof that apoptosis plays a role here. Do the authors detect enhanced cell death in the intestines of their mice? For instance, is TUNEL staining increased in sections of the gut? Although the stainings have been performed, I am missing a description of these results.

m) We have now quantified TUNEL positive cells in the intestine and found that there is an increased number of TUNEL positive cells in *Ripk1* ^{Δ CD4} mice (**Fig. EV2E**), the positive cells are mostly located in the epithelium (**Fig. 2G**).

On page 13/14 the authors discuss the role of TNF in gut inflammation. Could TNF play a role here in their system?

n) We observed an increased concentration of TNF in tissue samples of the small intestine in *Ripk1* ^{Δ CD4} mice (**Fig. 1G and Fig 7J**). Additionally, we found that especially intestinal $\gamma\delta$ T cells displayed a more inflammatory phenotype with increased TNF production (**Fig. 3K-L**). Villus atrophy in Crohn's disease has been associated with TNF production by IELs. Therefore, we crossed the *Ripk1* ^{Δ CD4} mice with *Tnfr1*^{-/-} knockout mice and found that although the intestinal elongation remained, the villus atrophy was blocked in the *Ripk1* ^{Δ CD4}*Tnfr1*^{-/-} mice (**Figs. 7 K-M**). This finding demonstrates a pathological role of TNF in the intestine of *Ripk1* ^{Δ CD4} mice while also showing that TNF is not responsible for the entire pathology. This complex pathology is an important finding when comparing to human IBD pathology and the varying response to TNF blockade (PMID: 39438660).

According to the scRNAseq data, Bcl2 expression is reduced in the conditional knockout mice. However, this applies to both, TCR β + CD8+ and TCR $\gamma\delta$ + populations, which are reduced and expanded, respectively. Is Bcl2 really the important molecule here? What about other cell death-related genes?

o) We agree with the reviewers assessment and have made new figures based on the scRNAseq data showing the gene expression cell death genes, including all mouse caspases, intrinsic apoptosis genes, and pore forming genes in the different T cell subsets, epithelial

cells and fibroblasts (Figs. EV3D and EV3H). We found that of the T cell subsets, only CD4 T cells had increased *Casp8* expression, underscoring CD4 T cells importance for the phenotypic rescue observed in the *Ripk1^{ΔCD4}Casp8^{ΔCD4}* mice. We have removed the text focusing on *Bcl2* regulation.

The expansion of $\gamma\delta$ T cells in the conditional knockout mice is impressive. However, the question arises what could be the driver of this $\gamma\delta$ T cell expansion? After all, *Ripk1* deletion happens in conventional T cells due to CD4-Cre driver. What is the authors idea on this phenotype?

p) We indeed observe enhanced $\gamma\delta$ T cell numbers in all organs that were investigated (Figs. 3 B-D). By use of the mixed bone marrow chimeras we show that $\text{TCR}\beta^+$ T cells have an intrinsic survival disadvantage in *Ripk1^{ΔCD4}* mice which is not shared by the $\gamma\delta$ T cells because they still express RIPK1 (Figs. 3 F-G and EV5H). Therefore, the increase $\gamma\delta$ T cell numbers is likely a niche-filling effect due to the lymphopenia of the conventional T cells. The consequences of this has been expanded upon in the discussion.

*Lines 401-404: “Furthermore, $\text{TCR}\gamma\delta^+$ T cell numbers are elevated in all examined peripheral tissues in *Ripk1^{ΔCD4}* mice. This increase in $\text{TCR}\gamma\delta^+$ T cells may reflect a compensatory mechanism resulting in RIPK1-sufficient $\text{TCR}\gamma\delta^+$ T cells filling the niche instead of $\text{TCR}\beta^+$ T cells due to the survival disadvantage of *Ripk1^{ΔCD4}* $\text{TCR}\beta^+$ T cells observed in bone marrow chimeras.”*

The antibiotics treatment strongly indicates a crucial role of the microbiota in the observed phenotype. However, there is a hen-and-egg problem: is the microbiota changed due to the ongoing inflammation or vice versa? Although addressing this question is clearly beyond the scope of the current study, the question remains as to why the intestinal microbiome is altered upon *Ripk1* deletion in T cells? Can the authors at least speculate how *Ripk1* deficiency in T cells impacts on the microbiome?

q) We have expanded on this in the following paragraphs:

*Lines 380-393: “In *Ripk1^{ΔCD4}* mice, certain features, such as crypt hyperplasia and intestinal elongation, were microbiota-dependent, whereas villus atrophy persisted despite broad-spectrum antibiotic treatment. The duodenal crypt hyperplasia, the intestinal elongation, and the expansion of Tregs, CD4^+ and $\text{CD8}\beta^+$ $\text{TCR}\beta^+$ IELs were all rescued following antibiotics, suggesting that these phenotypes represent a response towards the intestinal microbiome. The expansion of $\text{TCR}\gamma\delta^+$ IELs and the villus atrophy remained in antibiotics-treated *Ripk1^{ΔCD4}* mice, indicating that these symptoms were independent of the microbiome alterations. Comparing germ-free and specific pathogen-free (SPF) mice has also previously shown that the total $\text{TCR}\gamma\delta^+$ IEL numbers are relatively unaffected by the presence of the microbiota (Bandeira et al., 1990; Hoytema van Konijnenburg et al., 2017). $\text{TCR}\gamma\delta^+$ IELs may affect the microbial composition through the expression of antimicrobial peptides (Rezende et al., 2023). However, since the antibiotic treatment did not affect the villus atrophy or increased mortality in the *Ripk1^{ΔCD4}* mice, the microbiota is unlikely to be the underlying cause of these pathologies.”*

Although the our data indicate that the microbiota only affects part of the pathology, we did perform some microbiome analyses. Although there seem to be a higher variances in the alpha diversity of the *Ripk1^{ΔCD4}* samples, the data did not show significant differences between genotypes for any part of the intestine. These data have been included below for the reviewer's information, but are not included in the manuscript (**Rebuttal Figure 2**).

Type 3 immune response not necessarily anti-bacterial. Its primary target are extracellular pathogens such as certain bacteria and fungi. This should be corrected.

r) This has been corrected:

Lines 304-305: *"Type 3 immunity is primarily an immune response towards extracellular bacteria and fungi (Annunziato et al., 2015)."*

Minor points:

Lane 380: A space is missing at "...T cells.While..."

Lane 505: A space is missing at "...2mM EDTA..."

Lane 580: A space is missing at "...3nM..."

s) These points have been corrected.

Figure 1H: the labeling K45A is missing.

Figures 2, S2, S6 and S7: Please indicate in which color CD45 and TUNEL are labeled. This is not indicated in the figure, the legends or the materials and methods section.

t) These items have been corrected.

Referee #3:

- You should be trying to help the work get published not necessarily in this journal but ultimately.
 - Don't criticize an experiment unless you can tell the authors how they could do it better. "If you just want to throw darts," he would say, "go to the pub."
 - Keep in mind that no one ever built a statue to a critic.
 - Try to act as a peer in the process of peer review.
- Science Signaling 2009 Michael Yaffe

Title: RIPK1 ablation results in T cell apoptosis driving small intestinal inflammation in mice

Manuscript # EMBOR-2023-57617V1

General Remarks:

This study examines the phenotype of mice with a RIPK1 deficiency in conventional T-cells. The authors show that the mice develop a chronic wasting syndrome and intestinal pathology that is primarily confined to the duodenum. The phenotype is not observed in RIPK1 kinase dead mice, and concomitant ablation of caspase-8 prevented the pathology. This clearly shows that the phenotype depends on the scaffold function of RIPK1 and that it works by preventing apoptosis. This mirrors the phenotype of human patients and a question that could be addressed in the discussion is why whole body dysfunction in humans seems to recapitulate this cell type specific deletion in mice?

u) We have included a section in the discussion mentioning the similarities between the pathologies observed in conditional T cell deletion of RIPK1 and Caspase-8 with human patients with mutations in these genes:

Lines 467-474: Although patients with RIPK1 deficiency display T cell abnormalities in the periphery (Sultan et al., 2022), the intestinal T cells in their pathology have yet to be thoroughly investigated. Although human RIPK1 deficiency affects the entire body, the reported duodenal pathology (Uchiyama et al., 2019) is similar to that observed in Ripk1^{ΔCD4} mice by specifically ablating Ripk1 T cells. This parallel pathology between humans and mice indicates a central role of T cells in mediating intestinal pathology in humans with RIPK1 deficiency. The finding indicates that patients with RIPK1 deficiency may benefit from bone marrow transplants to ameliorate systemic immune cell deficiencies.

Since there is a clear increase in the TCR $\gamma\delta$ T population in the Δ CD4 RIPK1 knock out mice, it would add to the insights from the manuscript if the authors could show that these cells either don't have RIPK1 expressed or are insensitive to TNF induced cell death, unlike their other T-cell counterparts.

v) Since most $\gamma\delta$ T cells do not express CD4 during ontogeny (PMID: 21681197), they will also not express CRE, resulting in RIPK1 not being excised in this T cell subset. We have added a western blot showing that RIPK1 is not deleted in the gdT cell fraction of the IELs (**Figs. EV5H**) and explicitly mentioned this point in the text.

The manuscript is well written and presented, the statistics properly described, good numbers of mice are analysed.

w) Thanks for this appreciation.

Specific Remarks

A quibble, but line 52, can a D325 mutation in RIPK1 properly be described as a gain of function? It loses the ability to be cleaved by caspase-8 and the mutant cells are now more sensitive to cell death, but one could therefore to my mind equally claim that it is a loss of function.

x) We agree with this assessment and have adapted the sentence accordingly:

Lines 68-70: “Besides RIPK1 deficiency, mutation of the caspase-8-specific cleavage site (Asp325) leads to autoinflammatory disorders due to enhanced RIPK1 kinase activity-mediated apoptosis and necroptosis (Lalaoui et al., 2019; Newton et al., 2019).”

Rebuttal figures:

Rebuttal Figure 1. Analysis of CD8α⁺β⁻ and CD8α⁺β⁺ expressing γδT cells in the IEL fraction. Gray (Ripk1^{FL/FL}) and red (Ripk1^{ΔCD4}).

Rebuttal Figure 2. Microbiome analysis in small intestine and colon between co-housed *Ripk1^{ΔCD4}* mice and littermate controls. (A) Relative abundance at phylum level. (B) Relative abundance at genus level. (C) Alpha diversity: Richness, Evenness, Shannon diversity. No significant differences found between *Ripk1^{ΔCD4}* mice and littermate controls for any part of the intestine.

Dear Prof. Vandenabeele,

Thank you for the submission of your revised manuscript to our editorial offices. I have now received the report from the three referees that were asked to re-evaluate the study, you will find below. As you will see, the referees now fully support the publication of the study in EMBO reports.

Before I can proceed with formal acceptance, I have these editorial requests I ask you to address in a final revised manuscript:

- Please remove the running title from the manuscript title page.
- Please restrict the number of keywords to 5 and order the sections like this, using these names:
Title page - Abstract - Keywords - Introduction - Results - Discussion - Methods - Data availability section - Acknowledgements (including the funding information) - Disclosure and Competing Interests Statement - References - Figure legends - Expanded View Figure legends
- Thus, please move the funding information to the Acknowledgements.
- Please make sure that all the funding information is also entered into the online submission system and that it is complete and similar to the one in the acknowledgement section of the manuscript text file.
- We now use CRediT to specify the contributions of each author in the journal submission system. CRediT replaces the author contribution section. Please use the free text box to provide more detailed descriptions and do NOT provide your final manuscript text file with an author contributions section. See also our guide to authors:
<https://www.embopress.org/page/journal/14693178/authorguide#authorshippinguidelines>
- The Data availability section (DAS) is restricted for information on primary datasets produced in a study (e.g. RNA-seq, ChIP-seq, structural and array data) that are deposited in a public database. Please remove all further text not related to externally deposited datasets from this section. Moreover, please remove now referee access information, but keep the direct link to the dataset. Finally, please make sure the dataset is public latest upon online publication of the manuscript.
- Please use our reference format (we need et al. for citations with more than 20 authors):
<http://www.embopress.org/page/journal/14693178/authorguide#referencesformat>
- Please make sure that all figure panels are called out separately and sequentially. Presently, there are callouts for Figure S1 and S2. Please update these callouts (EV1 and EV2?). Please check.
- Please upload a complete author checklist with selected responses in the second column.
- Please check again that the number "n" for how many independent experiments were performed, their nature (biological versus technical replicates), the bars and error bars (e.g. SEM, SD) and the test used to calculate p-values is indicated in the respective figure legends. Please also check that all the p-values are explained in the legend, and that these fit to those shown in the figure. Please provide statistical testing where applicable. Please avoid the phrase 'independent experiment', but clearly state if these were biological or technical replicates. Please also indicate (e.g. with n.s.) if testing was performed, but the differences are not significant. In case n=2, please show the data as separate datapoints without error bars and statistics. See also:
<http://www.embopress.org/page/journal/14693178/authorguide#statisticalanalysis>
- If n<5, please show single datapoints for diagrams. Moreover:
 - Please note that the figure 5C is mislabeled as figure 5B in the legends of the manuscript. This needs to be rectified.
 - Please provide exact p values in the legends of figures 1B, C, D, F, G; 2B, C, F, H; 3A-D, F-H, J, K, L; 4B, E; 6B, C, G, H, I; 7D, J, K, L, M; EV1 B; EV2 E, G, H, I, J; EV4 A, D, E; EV5 E, I, J, K.
 - Please indicate the statistical test used for data analysis in the legends of figures 4C, D, F, G; 5B, C; EV3 B, C, F, G.
 - Please note that in figures 4B, E; EV1 B; EV2 E, G, H, I, J; there is a mismatch between the annotated p values in the figure legend and the annotated p values in the figure file that should be corrected.
 - Please add information related to n to the legends of figures EV1 H; EV2 E
 - Please define the error bars in the legend of figure EV1 H.
- Please provide titles for the EV figures in the EV Figure Legend.
- Please name Figure S2 Figure EV2 in the figure itself.
- Please add scale bars of similar style and thickness to microscopic images, using clearly visible black or white bars (depending

on the background). Please place these in the lower right corner of the images themselves. Please do not write on or near the bars in the image but define the size in the respective figure legend. Presently, most of the scale bars are rather small and all have text nearby.

- During our figure integrity check, we noted a partial overlap between Fig. 2G 'CD45-Tune1' (upper panel) and Figure EV5C (leftmost panel). See also the attached report file (57617V2-reuse.pdf). Please check. In case the reuse of the image is intentional, please state this clearly in the respective figure legends.

In addition, I would need from you uploaded separately:

Best,

Referee #1:

The authors have responded to all of my concerns in a satisfying manner. Overall, the revised manuscript substantially improved.

Referee #2:

The authors have addressed all of my previous concerns. I congratulate them to this fine study!

Referee #3:

The authors have responded well to my comments (which were not extensive anyway).

All editorial and formatting issues were resolved by the authors.

Prof. Peter Vandenabeele
VIB-UGent IRC
Technologiepark 71
Gent 9052
Belgium

Dear Prof. Vandenabeele,

I am very pleased to accept your manuscript for publication in the next available issue of EMBO reports. Thank you for your contribution to our journal.

Yours sincerely,
